# Cloud Aerosol Transport System (CATS) 1064 nm Calibration and Validation

Rebecca M. Pauly[1], John E. Yorks[2], Dennis L. Hlavka[1], Matthew J. McGill[2], Vassilis Amiridis[3], Stephen P. Palm[1], Sharon D. Rodier[4], Mark A. Vaughan[5], Patrick A. Selmer[1], Andrew W. Kupchock[1], Holger Baars[6], Anna Gialitaki[3]

[1]Science Systems and Applications Inc., Lanham, 20706, United States
[2]NASA Godard Space Flight Center, Greenbelt, 20771, United States
[3]National Observatory of Athens, Institute for Astronomy, Astrophysics, Space Application and Remote Sensing, Athens, Greece
[4]Science Systems and Applications Inc., Hampton, 23666, United States
[5]NASA Langley Research Center, 23618, United States
[6]Leibniz Institute for Tropospheric Research (TROPOS), Leipzig, Germany

*Correspondence to*: Rebecca Pauly (rpauly90@gmail.com)

**Abstract.** The Cloud-Aerosol Transport System (CATS) lidar on board the International Space Station (ISS) operated from 10
February 2015 to 30 October 2017 providing range-resolved vertical backscatter profiles of Earth's atmosphere at 1064 and 532 nm. The CATS instrument design and ISS orbit lead to a higher 1064 nm signal-to-noise ratio than previous space-based lidars, allowing for direct atmospheric calibration of the 1064 nm signals. Nighttime CATS Version 3-00 data were calibrated by scaling the measured data to a model of the expected atmospheric backscatter between 22 and 26 km above mean sea level (AMSL). The CATS atmospheric model is constructed using molecular backscatter profiles derived from Modern-Era
Retrospective analysis for Research and Applications, Version 2 (MERRA-2) re-analysis data and aerosol scattering ratios measured by the Cloud-Aerosol Lidar with Orthogonal Polarization (CALIOP). The nighttime normalization altitude region was chosen to simultaneously minimize aerosol loading and variability within the CATS data frame, which extends from 28 km to –2 km AMSL. Daytime CATS Version 3-00 data were calibrated through comparisons with nighttime measurements of the layer integrated attenuated total backscatter (iATB) from strongly scattering, rapidly attenuating opaque cirrus clouds.

25        The CATS nighttime 1064 nm attenuated total backscatter (ATB) uncertainties for clouds and aerosols are primarily related to the uncertainties in the CATS nighttime calibration technique, which are estimated to be ~9%. Median CATS V3-00 1064 nm ATB relative uncertainty at night within cloud and aerosol layers is 7%, slightly lower than these calibration uncertainty estimates. CATS median daytime 1064 nm ATB relative uncertainty is 21% in cloud and aerosol layers, similar to the estimated 16-18% uncertainty in the CATS daytime cirrus cloud calibration transfer technique. Coincident daytime
comparisons between CATS and the Cloud Physics Lidar (CPL) during the CATS-CALIPSO Airborne Validation Experiment (CCAVE) project show good agreement in mean ATB profiles for clear-air regions. Eight nighttime comparisons between CATS and the Polly[XT] ground based lidars also show good agreement in clear-air regions between 3-12 km, with CATS having a mean ATB of 19.7 % lower than Polly[XT]. Agreement between the two instruments (~7%) is even better within an aerosol layer. Six-month comparisons of nighttime ATB values between CATS and the Cloud-Aerosol Lidar with Orthogonal Polarization
(CALIOP) also show that iATB comparisons of opaque cirrus clouds agree to within 19%. Overall, CATS has demonstrated that direct calibration of the 1064 nm channel is possible from a space based lidar using the atmospheric normalization technique.

# 1 Introduction

Lidar plays a crucial role in observing the Earth's atmosphere as it enhances our understanding of the roles clouds and aerosols play in the climate system by providing vertical profiles of backscatter coefficient and other optical properties. Lidar have been utilized to study the vertical distribution and injection heights of smoke plumes (e.g. McGill et al., 2003, Wang et al., 2013, Rajapakshe et al., 2017), properties and transport of mineral dust aerosols (e.g. Papayannis et al., 2009, Yang et al., 2013, Haarig et al., 2017), and layer and optical properties of clouds (e.g. Yorks et al., 2011, Avery at al., 2012, Haarig et al., 2016, Noel et al., 2018). Lidar, particularly from a spaceborne platform, has the capability to provide these vertical profiles of cloud and aerosol optical properties globally.

To derive optical properties of clouds and aerosols from backscatter lidar systems, the signal must be accurately calibrated. While various methods have been used for calibrating lidar measured signal, the preferred method is the Rayleigh normalization technique, with minor if any corrections for aerosol contributions, as described in Russell et al. (1979). Ground based lidars (e.g. Micro-Pulse Lidar Network (MPLNet) (Welton et al., 2001)) calibrate by normalizing their signal to the molecular profile, but require knowledge of the aerosol optical depth of the atmosphere between the instrument and the calibration region (Welton et al., 2002). Since the MPLNet lidar sites are co-located with Aerosol Robotic Network (AERONET) (Holben et al., 1998) sites, the aerosol optical depth can be derived directly from the AERONET column optical depths measured by sun photometers.

High altitude airborne and spaceborne lidars have the benefit of weak aerosol loading in the atmosphere between the instrument and the calibration region. Spaceborne lidars (e.g. Lidar In-Space Technology Experiment (LITE) (Winker et al., 1996), the Geoscience Laser Altimeter System (GLAS) (Spinhirne et al., 2005), and the Cloud-Aerosol Lidar with Orthogonal Polarization (CALIOP) (Winker et al., 2010) have used a similar Rayleigh normalization technique to calibrate their 532 nm signals. Due to the weaker molecular signal to noise ratio (SNR) at 1064 nm compared to 532 nm for these instruments, calibration techniques for the 1064 nm attenuated total backscatter (ATB) calibration are based on the 532 nm ATB calibration (Vaughan et al., 2019).

Operationally, LITE did not calibrate its 1064 nm channel. GLAS and CALIOP use variants of the cirrus cloud calibration scheme proposed by Reagan et al. (2002). The CALIOP algorithms first calibrate the 532 nm data by normalizing the data between 36-39 km (Kar et al., 2018) to a modeled molecular density profile derived from the Modern-Era Retrospective analysis for Research and Applications, Version 2 (MERRA-2) re-analysis meteorological profiles (Gelaro et al., 2017). The 1064 nm signal is calibrated utilizing the 532 nm calibrated signal within cirrus clouds. Clouds are identified for use in the calibration algorithm based on thresholds applied to the magnitude of the 532 nm layer-integrated attenuated backscatter ($sr^{-1}$), cloud base and top altitudes, cloud temperature, and the layer-integrated 532 nm volume depolarization ratio (Vaughan et al., 2019). Using cirrus comprised of ice crystals assumed to be larger than the lidar wavelength ensures that the in-cloud backscatter coefficients at 1064 nm and 532 nm are essentially identical (Reagan et al., 2002, Vaughan et al., 2010, Haarig et al., 2016), thus enabling calculation of a 532-to-1064 calibration scale factor for each qualifying cirrus cloud identified in the CALIPSO backscatter data. These calibration scale factors are then composited into a continuous time history using a two-dimensional moving window averaging scheme that spans multiple orbits. For any individual profile, the CALIPSO 1064 nm calibration coefficient is simply the product of the interpolated instantaneous value of the scale factor time history and the corresponding 532 nm calibration coefficient (Vaughan et al., 2019).

The Cloud-Aerosol Transport System (CATS) (McGill et al. 2015) onboard the International Space Station (ISS) is unique in that its strong nighttime SNR at 1064 nm enables calibration of the 1064 nm nighttime data directly by normalizing the range corrected signal to a modeled molecular profile. There are three factors that enable the direct calibration of CATS 1064 nm data. First, CATS utilizes photon counting detectors that provide sufficient detection sensitivity at 1064nm (Yorks et al., 2016).

Second, the combination of low pulse energies (1-2 mJ) and higher repetition rate (4-5 kHz) lead to a higher output power (~8W) than all previous spaceborne lidars. Third, the CATS orbit on the ISS is considerably lower than other spaceborne lidars at ~405 km above mean sea level (AMSL). Section 2 of this paper discusses the CATS instrument, algorithms, and calibration. Section 3 discusses the uncertainties in the CATS calibration coefficients and attenuated total backscatter (ATB) measurements.

Comparisons with airborne, ground-based, and space-borne lidar are presented in Sect. 4. Concluding remarks are given in Sect. 5.

## 2 The CATS Instrument

CATS is an elastic backscatter lidar onboard the ISS, which operated nearly continuously from 10 February 2015 to 30 October 2017. With the ISS 51° inclination orbit, CATS provided diurnally varying measurements of clouds and aerosols. Over the course of the CATS lifetime, it operated in two modes. The first, mode 7.1, featured two fields-of-view with backscatter and depolarization information at both 1064 nm and 532 nm. Mode 7.1 utilized laser 1, which had a repetition ratio of 5 kHz and an output energy of ~1 mJ/pulse at both wavelengths. CATS operated in mode 7.1 for only 40 days due to a failure in laser 1

electronics, after which operations switched to mode 7.2. Mode 7.2 featured a single field of view, backscatter profiles at 1064 and 532 nm, and depolarization measurements at 1064 nm. Mode 7.2 used the second laser, which had a repetition ratio of 4 kHz and an output energy of ~2 mJ/pulse at 1064 nm. The different laser repetition rates yielded signal folding windows (see Section 2.1 for more details) of 30 km (mode 7.1) and 37.5 km (mode 7.2). To comply with ISS data rate limitations and simplify data system designs, the CATS data frame was set to -2.0 to 28 km (the lower of the signal folding windows of the two modes) for all

modes. CATS data is reported at a vertical sampling interval of 60 m for both modes, with a temporal resolution of 20 Hz (~350 m horizontal given the speed of the ISS), which required onboard integration of 200 laser shots in mode 7.2 (250 for mode 7.1). Since the majority of the CATS data was collected in mode 7.2, this paper primarily focuses on results from mode 7.2, although the calibration process is the same for both.

CATS Version 3-00 data products, which are the focus of this paper, consist of two primary data processing levels. To

create Level 1 (L1) data products, the raw CATS signal is range corrected, geolocated, corrected for detector non-linearity, and normalized to laser energy (measured onboard and averaged/reported at 20 Hz), producing the normalized relative backscatter (NRB). The NRB, in units of $km^2 J^{-1}$ counts, can be defined as:

$$NRB(r) = \frac{\{[N_s(r)*D]-N_B\}r^2}{E},\tag{1}$$

where r is the range (meters), $N_s$ is the geolocated CATS signal (photon counts), D is the correction term for detector non-

linearity (unitless), and E is the laser energy (Joules). Since the detectors employed by CATS have a deadtime of 28 to 30 ns for a discriminator maximum count rate on the order of 30 MHz, and CATS has a photon count rate of less than 35 MHz 99% of the time below 28 km, D is less than 1.10 for most atmospheric profiles (Yorks et al., 2015). $N_B$ is the photon counts from solar background, which can be determined by averaging the background signal acquired after the laser signal attenuated by Earth's surface and after the correction for the signal folding. Next, the signal is calibrated using the molecular profile derived from

MERRA-2 meteorological re-analysis data. The calibration coefficients, determined through the methods described below, can be found in each CATS L1B data file (also called granule). For Level 2 (L2) data products, aerosol and cloud layers are detected and optical properties are determined. Descriptions of the L2 algorithms are beyond the scope of this paper, but more information about both L1 and L2 processing algorithms can be found in the CATS Algorithm Theoretical Basis Document (ATBD) (Yorks et al., 2015).


## 2.1 CATS Nighttime Calibration

CATS exhibits high nighttime 1064 nm SNR, enabling 1064 nm attenuated total backscatter (ATB) direct calibration by normalizing the CATS signal to the Rayleigh profile corrected for aerosol contributions. Fig. 1 shows the CATS 1064 nm SNR for both night and day as compared to those of CALIOP 1064 nm. The CATS nighttime SNR is approximately an order of magnitude higher than that of CALIOP throughout the measurement column. On the other hand, the daytime CATS SNR is approximately a factor of 2 lower than CALIOP's, necessitating a different calibration technique for daytime data, as described in Section 2.2. The CATS nighttime signal is calibrated in the region between 22-26 km AMSL. There are two factors that determined this altitude region: (1) the CATS data frame is -2 to 28 km AMSL because the CATS laser 1 repetition rate of 5 kHz creates a 30 km atmospheric window for scattering from a single laser shot, and (2) testing of the highest possible altitude regions (based on #1) showed better performance in the 22-26 km than the 23-27 km region. While this altitude region provides sufficient molecular scattering for the Rayleigh normalization technique, the aerosol loading in the lower stratosphere (22-26 km) is also higher than the 36-39 km region used to calibrate 532 nm CALIOP data. To improve the accuracy of the CATS nighttime calibration, the aerosol loading in the calibration region must be quantified, along with the ozone transmission profile, molecular backscatter profile, and polarization gain ratio (PGR). Additionally, the background signal must be removed from the data. Since the CATS data is normalized to the Rayleigh profile corrected for aerosol contributions, more so than previous projects that have employed a similar technique, we will refer to the CATS nighttime calibration technique as the "atmospheric normalization technique" in this paper.

The nighttime atmospheric normalization technique is complicated by molecular folding of the raw signal caused by CATS' high repetition rate laser. Molecular folding refers to the fact that the CATS raw photon count at altitude, z, where z< 28 km, has scattering contributions from the atmosphere at heights z+Nx. N=1,2,3, etc., where x equals 37.5 km for mode 7.2 since laser 2 had a repetition rate of 4 kHz. The implications of this are that the region below the surface return (from -2.0 to 0.0 km), which is used for determining the background signal, also has molecular signal from 37.5 to 39.5 km. If this folded signal is not removed from the background signal, most of the signal in the calibration region will be removed by the background removal process. A correction term was implemented to account for this molecular folding. The folded signal is computed from instrument parameters and the known molecular attenuated backscatter cross section between 37.5 km and 39.5 km and subtracted from the signal in the background region (0.0 to -2.0 km below the ground). For nighttime data, this can affect the profile slope of the average signal above 20 km. If too much folding is removed, the slope will be greater than the molecular slope and if too little is removed, the average signal slope will be less than the molecular slope. In the data processing, a scaling factor in the folding equation is adjusted until the slope difference is less than 3.5%. The potential error introduced by this correction is discussed further in Sect. 3. For more information about molecular folding corrections, see the CATS ATBD (Yorks et al., 2015).

Depending on the profile location of the calibration region, the aerosol loading at those altitudes can introduce uncertainties in the computation of the calibration coefficient of any lidar system (Powell et al., 2009, Vernier et al., 2009, Kar et al., 2018). Thus, the CATS algorithm improves the calibration accuracy by incorporating a range-dependent particulate scattering ratio (unitless fraction), $R_\lambda(r) = \dfrac{\beta_m(r) + \beta_{p,\lambda}(r)}{\beta_{m,\lambda}(r)} = 1 + \dfrac{\beta_{p,\lambda}(r)}{\beta_{m,\lambda}(r)}$, where $\lambda$ indicates the wavelength of the measurement and $\beta_{m,\lambda}(r)$

and $\beta_{p,\lambda}(r)$ are, respectively, the volume backscatter coefficients for molecules and particulates (units $km^{-1}$ $sr^{-1}$) at range r, with particulates being understood to represent either cloud or aerosol particles. No space-based sensors provide stratospheric particulate scattering ratios at 1064 nm on a global scale. However, since robust estimates of the 532 nm scattering ratios in the CATS vertical

calibration zone can be readily derived from the CALIOP V4 Level 1 data, the CALIOP data is used to estimate the spatially and temporally varying 1064 nm scattering ratio at these altitudes (Fig. 2). Every 15 days, the CATS team computed 30-day zonal averages of the CALIOP 532 nm scattering ratios between 22 and 26 km. Given an estimate of the particulate (i.e., aerosol) backscatter color ratio (unitless fraction), $\chi_p = \dfrac{\beta_{p,1064}(r)}{\beta_{p,532}(r)}$, 

$$\chi_p = \frac{\beta_{p,1064}(r)}{\beta_{p,532}(r)}, \tag{2}$$

1064 nm scattering ratios can then computed from the 532 nm scattering ratios, which have been interpolated to the CATS vertical resolution, using

$$R_{1064}(r) = 1 + \frac{\chi_p\, \beta_{m,532}(r)\,\left(R_{532}(r)-1\right)}{\beta_{m,1064}(r)}, \tag{3}$$

where, following Hair $et\ al.$ (2008), $\chi_P = 0.40$ is taken as a constant for the aerosol loading in the upper troposphere/ lower stratosphere. This value is originally derived from data shown in Spinhirne et al. (1997). Sulfate aerosols are potentially the

largest contributor to the stratospheric aerosol loading (SPARC-ASAP, 2006; Vernier et al., 2015; Kresmer et al., 2016), and this value is also consistent with lower tropospheric measurements of sulfate aerosols (Groß,et al., (2013). The ozone transmission, $T^2_o(r)$, is determined from the MERRA-2 ozone mass mixing ratios and meteorological profiles. The ozone transmission is calculated using

$$T_o^2(\lambda, r) = exp\left[-2c_o(\lambda) \int_H^r \varepsilon_o(r')dr'\right], \tag{4}$$

where $\varepsilon_O(r)$ is the column density of ozone and $c_O(\lambda)$ is the Chappius ozone absorption coefficient (in cm$^{-1}$) obtained from a lookup table found in Iqbal (1984). The 1064 nm ozone coefficient is ~0.0 cm$^{-1}$ leading to the ozone transmission at 1064 nm being 1.0 and negligible to the 1064 nm signal calibration.

The molecular backscatter coefficient is calculated using the relationship to atmospheric temperature and pressure (Collins and Russell,1976), with

$$\beta_M = \frac{p}{KT}(5.45x10^{-32})\left(\frac{\lambda}{550}\right)^{-4.09}, \tag{5}$$

where $T$ is temperature (K), $p$ is the atmospheric pressure (Pa), and $K$ is the Boltzmann constant (m$^2$ kg s$^{-2}$ K$^{-1}$). The atmospheric profiles of temperature and pressure are obtained from the MERRA-2 re-analysis data. The atmospheric profiles are interpolated to the 60 m vertical resolution of the CATS lidar backscatter data. The molecular extinction coefficient ($\sigma_M$, units km$^{-1}$) is determined though the relationship:

$$\sigma_M = \beta_M \left(\frac{8}{3}\right)\pi \tag{6}$$

The PGR, which is reported in the Level 1B data files as metadata, is required to account for relative gain between the CATS parallel and perpendicular channels in the receiver. The PGR is determined from the scattered solar background radiation ratio of the parallel-to-perpendicular channels from dense cirrus clouds following the methodology from Liu et al. (2004). It can be assumed that the difference in solar background counts between the two channels is negligible because scattered solar radiation

from dense ice clouds is unpolarized (Liou et al., 2000). The CATS PGR is computed through the ratio of the sum of all parallel and perpendicular profiles in a daytime granule containing dense ice clouds. The profiles with dense ice clouds used in this computation, which are similar to those used in the CALIOP 1064 nm calibration technique as outlined in Vaughan et al. (2010), are identified through the following criteria:

1) Mid-cloud temperature < -35 C, as reported by MERRA-2

2) Cloud layer integrated ATB (iATB): 0.008 < iATB < 0.044 sr$^{-1}$

3) Layer integrated depolarization ratio NRB data ($\delta_{1064}$): $0.3 < \delta_{1064} < 0.8$

where:

$$\delta_{1064} = \frac{\sum_{layer} NRB_{perp} PGR_{hist}}{\sum_{layer} NRB_{par}} \text{ (where PGR}_{hist} \text{ is a historical PGR value)} \tag{7}$$

4) Cloud optical depth > 1.75 (estimated using the iATB and assumed lidar ratio of 25 sr)

These criteria were only used to identify cirrus clouds that would be suitable for calculating the PGR. Historical calibration coefficients and PGR values were used to estimate iATB, depolarization ratio, and optical depth. These historical values were not applied to the raw data during the actual PGR calculation. Because the CATS instrument ceased operation prior to the processing of CATS V3-00 data, a singular yearly average PGR value was used for 2015, 2016, and 2017 equaling 0.9839, 0.9768, and 0.9708 respectively. The PGR is applied as a multiplicative factor to the perpendicular channel NRB data. The perpendicular (multiplied by the PGR) and parallel NRB data are added together to arrive at the total NRB.

A single calibration coefficient for nighttime data is applied to the NRB profile on a per file, or granule, basis, using the methodology as follows to obtain the attenuated total backscatter ($km^{-1} sr^{-1}$) profiles. To prepare for calibration, the CATS night granules are separated into six segments averaging 7.8 minutes each, depending on the length of the granule. Granules are the files for the CATS data that span about half of the ISS orbit and contain only daytime or only nighttime observations. For calibration, the total NRB profile is averaged within each segment. The average total NRB profile is divided by the ozone transmission and scattering ratio of the corresponding wavelength as a function of height. The profiles of calibration coefficient ($C$), in units of $km^3 sr\ J^{-1}$ counts, for each segment within a file are determined by normalizing the mean NRB signal which has been corrected for aerosol loading and the ozone transmission, $\beta_{CN}$, to the mean molecular backscatter ($\beta_M T^2_M$) (Russell et al. 1979, Del Guasta 1998, McGill et al. 2007, Powell et al. 2009), via

$$C_\lambda(r) = \frac{\left[\frac{NRB(r)}{T^2_O(r)R(r)}\right]}{\beta_M(r)T^2_M(r)} = \frac{\beta_{CN}(r)}{\beta_M(r)T^2_M(r)}, \tag{8}$$

The final calibration coefficient for the segment is the average coefficient in the calibration region profile (i.e. an average of C(r) from 22 to 26 km). Each coefficient is compared to minimum and maximum threshold values, which vary based on the fluctuations shown in Fig. 3, to determine if the calculated value is within acceptable bounds. If the coefficient is not, it is discarded, and not used in the final calibration calculation. The calibration thresholds were determined through prior experience calibrating airborne lidar as well as through testing on CATS data during which outliers that negatively impacted the total calibration were identified. All good calibration values within a file are then averaged. On average, 67% of calibration values within a given granule are accepted and used for determining the final calibration coefficient for that file. If less than 15% of calibration values are accepted, a default calibration coefficient is used for that granule, computed as the mean of the calibration coefficients from the previous week of data. These files represent 3% of CATS data, typically when the laser was recently turned on after being off for more than 2 hours, and are noted in the Quality Control Flag variable in the CATS L1B data products. The final calibration coefficient, which is also reported in the Level 1B data files, is then applied to all NRB profiles within the granule to compute the ATB.

The time evolution of the CATS nighttime calibration coefficients is correlated with the thermal stability of the cooling loop on the ISS, which in turn is attributed to the changing of the sun's angle with respect to the ISS orbital plane, known as its beta angle. The CATS nighttime calibration coefficients oscillate from $4\times10^8$ to $1.4\times10^9$ $km^3 sr\ J^{-1}$ counts with a period of roughly 30-40 days. This oscillation is a result of changes in the CATS laser properties (i.e. wavelength, alignment, etc.) due to thermal instability of the cooling loop. The thermal instability of the cooling loop and instrument was monitored by the cold plate temperature. Fig. 3 shows the daily average nighttime calibration coefficient (black x's and black line) and CATS cold plate

temperature (blue) for the entire mode 7.2 dataset (top, April 2015 – October 2017) and for a subset from January- April 2016 (bottom). The changing value of the calculated calibration coefficient follows the same pattern as the cold plate temperature. The daily average nighttime calibration coefficient and cold plate temperature have a correlation coefficient of 0.8066 during the period of January- April 2016.

## 2.2 CATS Daytime Calibration

Because CATS daytime data exhibits lower SNR due to solar background noise, calibrating the daytime granules through the atmospheric normalization method is not possible. Therefore, the daytime calibration coefficients are determined through calibration transfer from the nighttime calibration (Eq. 9). In previous CATS data versions, the daytime calibration was determined through a manual normalization to the Rayleigh profile corrected for aerosol contributions that required periodic assessment and updates. For V3-00 CATS data, a single daytime calibration coefficient was determined for each calendar month of CATS data through an assessment of the iATB ($sr^{-1}$) in strongly scattering opaque cirrus clouds that have a mid-layer temperature colder than -20° C (based on the MERRA-2 reanalysis data) and a layer integrated depolarization ratio between 0.25 and 0.7. Only highly scattering, rapidly attenuating clouds (CATS signal attenuated in 2 km or less) were used in the assessment.

It was found that using a month of data provided enough data points to compute a calibration value while also reasonably capturing the temporal variability of calibration coefficients. The assessment of cirrus cloud properties was done using V2-01 CATS data in which the layer detection and optical properties algorithms were already run. A layer is classified as opaque if no layer or ground signal is detected below it. The iATB is calculated through the cloud until the point of signal attenuation. For the strongly scattering, rapidly attenuating opaque cirrus selected for the daytime calibration transfer procedure, there should be little difference between nighttime and daytime iATB retrievals. This characteristic of cirrus clouds has been observed in CALIOP data as shown in Young et al. (2018). Young et al.'s CALIOP comparisons of opaque cirrus at 532 nm showed substantial iATB similarities for both nighttime and daytime measurements, with a peak iATB of ~0.03 $sr^{-1}$ in both cases. Given that there is relatively little difference in the backscatter from cirrus clouds between 532 nm and 1064 nm (Vaughan et al., 2010, Haarig et al., 2016), one would expect that the daytime and nighttime iATB distributions from 1064 nm retrievals should also be similar.

The daytime calibration coefficient is computed as

$$C_{day} = \frac{\frac{1}{N_{day}}\sum_{k=1}^{N_{day}} iNRB_k}{\frac{1}{N_{night}}\sum_{k=1}^{N_{night}} iATB_k}, \qquad (9)$$

where both the nighttime iATB and daytime iNRB were computed over each calendar month of CATS data. The left panel of Fig. 4 demonstrates the CATS daytime calibration for the month of August 2016. In the CATS V2-01 data, the daytime cirrus iATB distribution is shifted higher than the nighttime distribution, with a peak at 0.05 $sr^{-1}$. For the V3-00 CATS processing, the daytime calibration coefficient for August 2016 was increased from $6x10^8$ to $9x10^8$ $km^3sr$ $J^{-1}$ counts and was applied to all August 2016 daytime granules. As seen in the right panel of Fig. 4, this change resulted in the peak of the daytime cirrus iATB distribution moving to ~0.03 $km^{-1}sr^{-1}$ with better agreement with the nighttime distribution. Overall, it was found that a change of ~$1x10^8$ $km^3sr$ $J^{-1}$ counts in the calibration coefficient results in a shift of ~0.01 $sr^{-1}$ in the iATB. This method was applied to all CATS mode 7.2 daytime data in V3-00 on a monthly basis. Changes in the nighttime cirrus iATB distributions between versions are attributed to improvements in the layer type classifications within the L2 processing (Yorks et al., in prep.).

Since the CATS daytime calibration coefficient is directly related to the nighttime calibration coefficient, the evolution of the daytime calibration coefficients are also correlated to the thermal stability of the cooling loop on the ISS (red dots, Fig. 3).

Most of the CATS daytime calibration coefficients range from $6\times10^8$ to $9.0\times10^8$ km$^3$sr J$^{-1}$ counts, with less variability compared to the nighttime calibration values given they are monthly-mean values (less temporal resolution than the nighttime values since it is computed every month and not every granule). The loss of this temporal resolution of the daytime calibration coefficients introduces a bias compared to the nighttime calibration coefficients. Overall, the daytime calibration method results in average biases of roughly 10%, based on the mean, median, and mode daytime iATB values with respect to nighttime of 0.000168, -001215, and -0.00258 sr$^{-1}$, respectively (Table 1). The mean absolute error (MAE) values also indicate that, overall, the distribution statistics between night and day granules are similar, with MAE values equating to 8-13% error in the peak of the distribution and 17% error in the standard deviation of the distribution.

## 3 Error Analysis

There are two types of error that contribute to the uncertainty in the CATS calibration: systematic and random errors. There are four sources of uncertainty included in the systematic error calculation (Yorks et al., 2015). They are: uncertainties in the scattering ratios (R) at 22-26 km from CALIOP, including assumptions of backscatter color ratio, uncertainties in the molecular backscatter ($\beta_M$) computed from MERRA-2 data, uncertainty in the modeled two-way transmittance (T$^2$) from atmospheric molecules and ozone, and errors introduced by the CATS optical system. The optical system error can be reduced through corrections such as deadtime correction and energy normalization to less than 0.1% and is therefore negligible. The total systematic error in the calibration, following the method outlined by Powell et al. (2009), can be defined as

$$\left(\frac{\Delta C}{C}\right)_{sys} = \sqrt{\left(\frac{\Delta R}{R}\right)^2 + \left(\frac{\Delta \beta_M}{\beta_M}\right)^2 + \left(\frac{\Delta T_M^2}{T_M^2}\right)^2 + \left(\frac{\Delta \chi_p}{\chi_p}\right)^2} \tag{10}$$

The errors in the molecular backscatter and background transmission are assumed to be constant, equaling 3% and 0.2% respectively. Regan et al. (2002) estimates transmission uncertainty for the 532 nm molecular backscatter coefficient of 3% and uncertainty at 1064 nm at a nominal cirrus cloud top altitude of 0.2%. Thus, the constant 3% molecular backscatter uncertainty is conservative, and results from uncertainties in GMAO-derived temperatures from the upper troposphere that are estimated to be less than 1 °C (Campbell et al., 2015). Omitting the King factor, which accounts for the anisotropy of molecules, in our molecular backscatter computation leads to an additional error of 1-3% in the molecular backscatter error (Hostetler et al., 2005). This additional error contributes to less than 1% error in the total systematic calibration, making is far less important than other factors covered in this paper, especially given that the 1064 nm molecular backscatter uncertainty is likely overestimated. The error in the scattering ratio is dominated by the uncertainty and variability of the CALIOP nighttime scattering ratios, which ultimately results from the uncertainty in the CALIOP nighttime calibration, estimated to be 1.6% ± 2.4 (Kar et al., 2018). The final source of systematic error is the assumption that the backscatter color ratio of the stratospheric aerosols between 22 and 26 km is constant at 0.40. To the authors' knowledge, the variability of the stratospheric aerosol backscatter color ratio is not documented in the literature, but an analysis of Stratospheric Aerosol and Gas Experiment (SAGE) III extinction Angstrom exponent averaged from June 2017 to August 2018 in the CATS calibration region yields 1.79 ± 0.10. Thus, we assume an absolute uncertainty in the stratospheric aerosol backscatter color ratio of 0.024. Applying Eq. (10) to these values, the total systematic relative uncertainty in the CATS calibration coefficients is estimated at 7%.

The random error in the CATS calibration is primarily caused by noise in the lidar signal during the calibration normalization. The random error can be determined through the variability of the NRB signal within each calibration segment (Welton and Campbell, 2002) and is calculated through

$$\left(\frac{\Delta C}{C}\right)^2_{ran} = \left(\frac{\frac{stdev(NRB(r))}{\sqrt{N}}}{NRB(r)}\right)^2,$$ (11)

where N is the total number of NRB values used. For CATS, the 7.8 min averaging interval equals 9,360 profiles. This averaging interval was chosen because it reduced the random error of each individual calibration value within a granule, but still provided sufficient values (at least 6) to compute the granule mean calibration coefficient. Uncertainties in background subtraction and other CATS correction terms (discussed in Sect. 2) are included in the NRB variability. The mean random error in the calibration coefficient is 6%. The total error is determined through

$$\left(\frac{\Delta C}{C}\right)^2_{tot} = \left(\frac{\Delta C}{C}\right)^2_{sys} + \left(\frac{\Delta C}{C}\right)^2_{ran}$$ (12)

and thus, comes to a total relative uncertainty in the CATS nighttime calibration ($\Delta C/C$) of ~9%.

The daytime calibration uncertainty can be estimated from the variability of the NRB signal and the nighttime calibration error. The nighttime calibration already contains several systematic uncertainties that are inherited during the calculation of the daytime calibration coefficients. Additionally, since strongly scattering cirrus clouds are used in the daytime calibration, uncertainties in the multiple scattering factor, η, should also be considered. Multiple scattering occurs when laser light emitted by the lidar interacts with more than a single particle within a scattering volume. Multiple scattering can lead to higher detected signals and is corrected using the appropriate value of η (Platt, 1979, Garnier et al., 2015). For CATS, η for cirrus clouds was determined to be 0.52 through comparisons with the Cloud Physics Lidar (CPL). The magnitude of multiple scattering contributions to the backscattered signal depends critically on both instrument viewing geometry and particle phase function. However, since neither of these factors is expected to show any discernable diurnal variability, we assume that uncertainties in our knowledge of η can be neglected when assessing the error sources for the CATS daytime calibration. Since a constant daytime calibration coefficient is determined for each month of CATS data and is based on comparisons with nighttime data, the total systematic error for the daytime calibration can be estimated to be the same as the average nighttime calibration uncertainty over the month.

The daytime random error is estimated from the variability in the NRB signal. Therefore, the total daytime error can be shown through the equation

$$\left(\frac{\Delta C}{C}\right)^2_{day} = \left(\frac{1}{N_{day}}\right)^2 \sum_{k=1}^{N_{day}} (\Delta iNRB)^2_{day,k} + \left(\frac{1}{N_{night}}\right)^2 \sum_{k=1}^{N_{night}} (\Delta iNRB)^2_{night,k} + \frac{1}{N_{night}} \sum_{k=1}^{N_{night}} (\Delta C)^2_{night,k}$$ (13)

The daytime random error due to the noise in the NRB is estimated to be ~15%, leading to a total daytime calibration uncertainty of ~16-18%.

The ATB uncertainties are computed using a propagation of errors from the NRB uncertainties. ATB is calculated through

$$ATB = \frac{NRB}{C}.$$ (14)

NRB uncertainties ($\Delta NRB$) are calculated using the methodology outlined in Welton and Campbell (2002). By utilizing a standard propagation of errors from the NRB uncertainty and the calibration uncertainty, the ATB uncertainty was computed and can be expressed as

$$\Delta ATB = \sqrt{\left(\frac{1}{C}\right)^2 \Delta NRB^2 + \Delta C^2 \left(\frac{NRB}{C^2}\right)^2}.$$ (15)

As part of the NRB error, there is error associated with the molecular folding correction factor (see Sect. 2.1) which impacts the ATB profile. Since the correction factor acts by matching the slope of the measured signal to that of the modeled

molecular profile within the calibration region, the error was assessed through the amount of error introduced lower in the CATS profile for given errors in the calibration region slope. In V3-00, the majority of corrected slopes in the calibration region have an error of less than 3.5%. However, in very few cases, the slope is different from the molecular slope by 10% in the calibration region. The assessment of this "worst case" calibration region slope error showed that the maximum error introduced in the profile is ~4% in the 17-18 km region. The error in the profile then decreases as the signal approaches the surface, introducing ~2% error.

The CATS 1064 nm ATB uncertainties for clouds and aerosols at night are primarily a related to the uncertainties in the atmospheric normalization technique. Features such as cloud and aerosol layers with higher backscatter intensities tend to have lower ATB uncertainty, while clear air regions, with lower scattering intensity and lower SNR, have higher ATB uncertainty. The median CATS 1064 nm ATB relative uncertainties from the Mode 7.2 V3-00 data products within cloud and aerosol layers are 7% at night and 21% during daytime. For clear-air regions, there is large variability (20% to over 100%) in the CATS 1064 nm ATB relative uncertainties, since the SNR varies as a function of altitude at night due to molecular scattering and scene during daytime due to the noise introduced from the solar background.

## 4 Data Comparisons

### 4.1 Airborne Lidar Comparisons

During the CATS-CALIOP Airborne Validation Experiment (CCAVE) in August 2015, the NASA ER-2 conducted several ISS under flights. As part of the CCAVE payload, CPL was able to collect coincident data with CATS. CPL is an airborne backscatter lidar that has participated in over thirty field campaigns, including several satellite instrument validation projects (McGill et al., 2002). CPL data products include ATB from both 1064 and 532 nm. Similar to nighttime CATS data, CPL is calibrated by normalizing the signals acquired between 15 km and 17 km to a modeled molecular attenuated backscatter profile derived from MERRA-2 reanalysis data. A 1064 nm particulate scattering ratio of 1.27 is applied in the calibration region for 1064 nm data, based on the work of Vaughan et al. (2010), and the estimated aerosol loading within a standard atmospheric profile in the northern hemisphere.

Fig. 5 shows the coincident flight from the CCAVE project which occurred at 01:37 UTC on 7-8 August 2015 during the day over western Nevada. CPL flew beneath CATS with clear sky conditions, although this scene is made more complicated due to variations in the terrain and background smoke aerosols due to wildfires in the region, as can be seen in the curtain plot (Fig. 5-left). The CATS average is comprised of 165 profiles which spans 55 km, and is calibrated using the daytime cirrus cloud calibration transfer technique. The CPL mean profile is an average of 280 profiles. Despite the complicated terrain and smoke, the mean ATB profiles from CATS and CPL still shows good agreement in the clear sky region above the smoke, with the average CPL and CATS mean ATB between 7-15 km equal to $4.1927 \times 10^{-5}$ [km$^{-1}$sr$^{-1}$] and $4.0972 \times 10^{-5}$ [km$^{-1}$sr$^{-1}$] respectively, meaning the CATS average ATB was 2.28% below CPL. This agreement is surprising since the CATS daytime calibration uncertainty is ~16-18%, but this case occurred near local twilight when CATS SNR is higher and the 1.27 value of the 1064 nm particulate scattering ratio used for CPL could be too low, introducing errors in the CPL 1064 nm ATB profile.

Another daytime underpass occurred at 20:31 UTC on 20 August 2015 over northern Utah near Great Salt Lake. The CPL curtain plot and the mean ATB profile from both CATS and CPL centered around the overpass time can be seen in Fig. 6. The CPL data was averaged to six minutes (360 ATB profiles) which covers a distance of about 70 km. The CATS data were averaged over the same distance and is comprised of 210 ATB profiles, and like the 7-8 August 2015 case is calibrated using the daytime cirrus cloud calibration transfer technique. As shown in the CPL curtain plot, the underpass segment was in clear-sky

conditions (no clouds) with a well-defined smoke aerosol layer from nearby wildfires. Both instruments observed the top of this aerosol layer around 5 km AMSL. The differences in SNR are also apparent as the CATS profile is noisier than the CPL profile. The average CPL ATB value between 7-15 km was $4.2967 \times 10^{-5}$ [km$^{-1}$sr$^{-1}$] and the average CATS ATB was $5.1939 \times 10^{-5}$ [km$^{-1}$sr$^{-1}$], 20.88% higher than CPL. These differences are expected given the 16-18% CATS daytime calibration uncertainty.

5        The greater noise in the CATS signal on the 20 August case should be noted as compared to the 7-8 August case. This is likely attributed to the different times of day the two flights occurred. The 7-8 August flight occurred in the early evening, which will minimize the noise induced by solar background due to the lower sun angles, while the 20 August flight occurred closer to local noon, which will maximize noise from sunlight. For both under-flights, the error in the CATS ATB compared with CPL is well within the uncertainty estimates of both instruments.

## 4.2 Ground-based Comparisons

       In addition to the coincident airborne CPL data, CATS was also compared to ground-based systems. CATS frequently passed over (or close to) the European Aerosol Research Lidar Network (EARLINET) sites. The Polly[XT] lidar (Baars et al.,
2016; Engelmann et al., 2016) is a Raman lidar developed by at the Leibniz Institute for Tropospheric Research (TROPOS), Leipzig, Germany and is used at some EARLINET sites. The Polly[XT] systems emit laser pulses at 1064, 532, and 355 nm with elastic backscatter detectors at each wavelength, as well as Raman channel detectors at 386.73 and 607.4 nm. There are Polly[XT] lidars all across Europe as part of the EARLINET, but only data collected from the Leipzig, Germany (51.3N; 12.4E) and the Athens NOA (National Observatory of Athens) (37.97 N; 23.71 E) sites were used in this study.

20        Raw EARLINET data are processed through the Single Calculus Chain (SCC) (D'Amico et al., 2015). The first part of the SCC is the EARLINET Lidar Pre-Processor (ELPP) where the raw lidar signal is range and deadtime corrected, the background signal is subtracted, and molecular extinction and transmission profiles are computed from meteorological radiosonde data or the standard atmosphere (D'Amico et al., 2016). The second part of the SCC is the EARLINET Lidar Data Analyzer (ELDA) (Mattis et al., 2016). In the ELDA the backscatter coefficients, extinction coefficients, and lidar ratio are
derived. During the backscatter coefficient calculation, the EARLINET data is calibrated by normalizing it to the molecular using an assumed aerosol free region, which is determined by the ELDA algorithms.

       Using the particulate backscatter and particulate extinction profiles derived from the Polly[XT] data, "CATS-like" ATB profiles were calculated following the methodology outlined in Mona et al. (2009) where the attenuated backscatter coefficient can be defined as

30        $\beta'(z) = \beta_{tot}(z) T_{par}^2(z) T_M^2(z).$                   (16)

$\beta_{tot}$ is the total backscatter coefficient comprised of contributions from particles, molecules, and ozone. $T_{par}^2$ is the particulate transmittance and is calculated through

       $T_{par}^2(z) = exp\left(-2 \int_z^{z_s} \alpha_{par}(\zeta) d\zeta\right),$            (17)

where $\alpha_{par}$ is the particulate extinction and $z_s$ is the CATS altitude. The particulate backscatter was computed from the Polly[XT]
1064 nm and 607 nm signals through the methodology described in Proestakis et al. (2019). The uncertainty in the backscatter coefficient retrieval is estimated to be between 5-20% (Ansmann et al., 1992; Whiteman et al., 2003; Povey et al., 2014). The particulate extinction coefficient was calculated using the Klett method (Klett, 1981; Fernald, 1984) using assumed lidar ratios between 30 - 35 sr. Sun photometer data was used, wherever possible, to estimate the lidar ratio. The molecular signal and attenuation profiles were computed from the temperature and pressure profiles found within the CATS L1B HDF5 file
corresponding to the overpass.

Fig. 7 shows the mean ATB profiles from the nighttime CATS overpass of the Leipzig Polly[XT] site on 24 September 2015 at 01:13:34 UTC. CATS passed 31 km from the Leipzig site. The mean profiles consist of forty CATS ATB profiles (~10 km) and thirty minutes of Polly[XT] data (36,000 profiles). This difference in number of averaged profiles is a contributing factor to the difference in the noise between the two instrument profiles. The CATS mean ATB profile was 7.7 % higher than the Polly[XT] mean CATS-like signal between 3-12 km. Another nighttime overpass, shown in Fig. 8, occurred on 30 July 2015 at 00:18:19 UTC ~41 km away from the Leipzig site. In this overpass, CATS ATB was 14.1% lower than the Polly[XT] data between 3-12 km.

Overall, eight clear-sky, nighttime overpasses were used in this analysis. The average difference from 3-12 km between CATS and Polly[XT] ATB was 19.7% with an average CATS distance from the Polly[XT] site of 40 km (Fig. 9). Fig. 9 also shows the CATS and Polly[XT] ATB scatter plot from all eight overpasses. The correlation coefficient between the two instrument retrievals is 0.75. The difference between the two instruments falls within the uncertainties in the CATS ATB (Sect. 3) and the uncertainties in the Polly[XT] retrievals. In addition to the clear sky comparisons, one overpass which had strong aerosol scattering within the planetary boundary layer (PBL) was assessed. The center-most 1.25 km of the PBL depth retrievals were compared to avoid spatial inhomogeneities in the PBL top and ground height. CATS underestimated Polly[XT] by 7%, supporting the ATB uncertainty assessment in Sect. 3 of lower ATB uncertainties (~8%) within stronger backscattering layers. Given the high SNR of CATS 1064 nm nighttime signal (Fig. 1), these differences can be primarily attributed to the ~9% uncertainty in the CATS nighttime atmospheric normalization calibration technique.

Previous studies have investigated the validity of using EARLINET for spaceborne lidar validation (Mamouri et al., 2009; Papagiannopoulos et al., 2016; Proestakis et al., 2019) and have found it is a useful method for lidar validation. A major source of the variability between the ground-based and spaceborne measurement results was found to be the variances in the atmospheric scene observed due to the spatial and temporal differences in the measurements. In a CALIOP validation study by Mamouri et al. (2009), it was found that for comparisons where the over pass was within 100 km from the EARLINET site the variability of the aerosol loading introduced a discrepancy on the order of 5%.

## 4.3 CALIOP Comparison

In addition to coincident data, statistical comparisons with CALIOP measurements can be used to further assess the CATS calibration. However, differences in instrument design can make the interpretation of these comparisons somewhat challenging. CALIOP measures the total backscatter in the 1064 nm channel using a single avalanche photodiode (APD), which simultaneously delivers a desirable high quantum efficiency and a less desirable high dark noise count rate that has been increasing linearly over the course of the mission (Hunt et al., 2009). CATS, on the other hand, uses a pair of photon counting modules to separately measure the 1064 nm backscatter components polarized parallel and perpendicular to the polarization plane of the CATS laser. The difference in detector performance is illustrated in Fig. 10, which shows the occurrence frequencies of the attenuated backscatter coefficients measured by CATS and CALIOP between 1 April and 30 September 2016 at all latitudes between 51.8° N and 51.8° S. This comparison was designed to investigate distributions of cirrus cloud backscatter intensity, so the data are restricted to nighttime measurements extending from 0 to 5 km above the point where the atmospheric temperature in any profile first drops below –40° C.

Because CATS uses photon counting detectors, the molecular backscatter signals in the CATS distribution appear as a sharp, well-confined peak at $\sim 2.5 \times 10^{-5}$ km$^{-1}$ sr$^{-1}$. The substantial broadening of the CALIOP distribution in this region is a consequence of the high APD dark noise levels in the CALIOP detectors. The distributions begin to converge above ~0.008 km$^-$

[1] sr[-1], although the CATS occurrence frequencies remain persistently lower than CALIOP throughout. Approximately 99.7% of all attenuated backscatter coefficients measured for both lidars lie below 0.025 km[-1] sr[-1]. Some of the differences at higher ATB values may be a consequence of the fact that these are not coincident measurements; because the two instruments fly in very different orbits, they sample different regions of the atmosphere at different times of day. CALIPSO flies in a sun-synchronous 98° orbit with a 16-day repeat time, and thus CALIOP measurements are acquired at approximately the same time of day at any given location along the orbit track (Hunt et al., 2009). The ISS flies in a 52° precessing orbit with a 3-day repeat time, so that CATS measurements at identical locations will occur at many different times of the day. This precessing orbit allows CATS data to be used to assess the diurnal variability of clouds and aerosols.

To avoid the confounding effects introduced by APD dark noise contamination of the weaker signals measured by CALIOP, a second study was conducted comparing the iATB from opaque cirrus detected by the two sensors. This study used CATS and CALIOP data acquired between 1 March and 31 December 2016, with the latitude range once again confined to between 51.8° N and 51.8° S. The following cloud selection criteria were applied uniformly to both data sets.

(a) All layers must be classified as opaque ice clouds and be the uppermost (and only) layer in the column.
(b) All layers must be detected at a nominal 5-km horizontal averaging resolution.
(c) The mid-layer temperature for all layers must lie below –37° C (see Campbell et al., 2015).
(d) Only nighttime measurements are used.

A comparison of the resulting frequency distributions is shown in Fig. 11. Descriptive statistics of the iATB values measured by each lidar are given in Table 2. In both Fig. 11 and Table 2, the mean CATS iATB is seen to underestimate CALIOP by ~11.8%. However, direct comparisons of mean iATB measured in opaque cirrus cannot fully characterize the calibration differences between the two instruments. In particular, any comprehensive evaluation must consider differences in the contributions of multiple scattering to the backscattered signals. Instrument-specific causes for these differences include different laser spot sizes, different receiver fields of view, and different orbit altitudes.

The iATB for opaque layers can be expressed in terms of the layer extinction-to-backscatter ratio, S (more commonly known as the lidar ratio), and a dimensionless, instrument-specific multiple scattering factor, η, using Platt's equation (Platt, 1973):

$$iATB = \frac{1}{2\eta S}. \tag{18}$$

Aggregating 10 months of nighttime measurements acquired within the same time frame and latitude limits yields very large sample sizes for both data sets, so we can reasonably assume that the distribution of lidar ratios observed by CATS and CALIOP are essentially identical. But we cannot assume that the CATS and CALIOP multiple scattering factors are identical, as they depend not only on the phase functions of the measurement targets (in this case, cirrus clouds) but also on instrument design and viewing geometry (Winker, 2003). As mentioned in Sect. 3, the value of η for CATS ($\eta_{CATS}$ = 0.52) has been determined empirically via comparisons to coincident CPL measurements. The cirrus multiple scattering factors applied in the CALIOP V4.10 data release were also determined empirically using extensive coincident measurements made by the CALIPSO infrared imaging radiometer (Garnier et al., 2015). Unlike CATS, $\eta_{CALIOP}$ is not a fixed constant, but is instead implemented as a function of cloud temperature.

For the opaque cirrus clouds sampled by CALIOP in this study, $\eta_{CALIOP} = 0.55 \pm 0.06$. Assuming that both instrument teams have accurately characterized cirrus multiple scattering effects on their respective systems, enforcing the assumption that the lidar ratio distributions observed by CATS and CALIOP are essentially identical, we can establish the relative difference in attenuated backscatter measurements between the two lidars using $(iATB_{CALIOP} \times \eta_{CALIOP}) / (iATB_{CATS} \times \eta_{CATS}) = (0.0313 \times 0.55)$ / $(0.0280 \times 0.52) = 1.182$. This result is consistent with the previous PollyXT comparisons. In "clear air" regions, the difference between CATS V3-00 L1B data products and PollyXT measurements of ATB is ~19.7%. In opaque cirrus, differences between CATS V3-00 L1B data products and CALIOP measurements of iATB is ~18.2%.

## 5 Conclusion

This study presents the CATS 1064 nm calibration algorithm, as well as validation using three different data sources. Cloud and aerosol layers have strong backscatter intensities and high SNR, so the CATS 1064 nm ATB uncertainties in these layers are primarily related to the uncertainties in the CATS calibration. At night, CATS V3-00 median 1064 nm ATB relative uncertainty is 7% in cloud and aerosol layers, slightly lower than the estimated ~9% uncertainty in the atmospheric normalization technique. The daytime cirrus cloud calibration transfer technique has an estimated uncertainty of 16-18%. CATS V3-00 median daytime 1064 nm ATB relative uncertainty is 21% in cloud and aerosol layers. Coincident flights with the airborne CPL instrument showed that even in conditions with peak solar background noise and lowest SNR, CATS data agrees to within 25% with CPL. The CATS ATB was also compared with the ground based EARLINET systems and found to be within 20% of the calibrated EARLINET data. Finally, CATS was compared in a statistical sense with CALIOP, another spaceborne lidar utilizing a different 1064 nm calibration method than CATS, and also found ATB agreement to ~18%. The comparisons between CATS, CPL, Polly[XT], and CALIOP 1064 nm ATB fall within the combined estimated uncertainties for the all the instruments. The results shown in this paper are critical to understanding the uncertainties in CATS 1064 nm Level 2 data products, as the calibration uncertainties from backscatter lidars generally impose a lower-bound on the uncertainties in cloud/aerosol extinction and optical depth retrievals from such instruments (Young et al. 2013; 2016). To date, the CATS cloud and aerosol top/base heights have been used for various applications, including volcanic plume transport (Hughes et al 2016), above cloud aerosol properties (Rajapakshe et al. 2017), pyrocumulonimbus smoke heights (Christian et al., 2019), and cloud diurnal variability (Noel et al. 2018). More recently, CATS cloud and aerosol optical properties (e.g. extinction, optical depth, ice water content) from Level 2 V-3.00 data have been compared to EARLINET aerosol products (Proestakis et al., 2019), used to estimate thin cirrus radiative forcing (Dolinar et al., 2019), and demonstrated the diurnal variability of aerosol properties (Lee et al. 2019).

CATS has demonstrated that direct calibration of 1064 nm from spaceborne lidar is possible given the appropriate instrument design and orbit. The CATS design and ISS orbit yielded data that exhibits high nighttime SNR, enabling the direct calibration of the nighttime CATS 1064 nm ATB by normalizing the signal to the Rayleigh profile corrected for aerosol contributions. The primary strength of this technique is that it does not require assumptions about cirrus cloud 1064-532 nm backscatter color ratios, as is the case with the CALIOP 1064 nm calibration technique (Vaughan et al., 2019). The accuracy of the atmospheric normalization technique, which is also used for CALIOP 532 nm data, is dependent on an accurate estimate of the aerosol loading in the calibration altitude region. A weakness of the CATS 1064 nm atmospheric normalization technique is that assumptions about the 1064-532 nm backscatter color ratio for stratospheric aerosols is used because accurate measurements of the 1064 nm aerosol loading in the 22-26 km altitude region, which has higher aerosol loading than the 36-39 km region used for CALIOP, were not available in same timeframe as CATS operations. To improve the calibration of future space-based lidar missions, especially at 1064 nm, a higher calibration altitude region should be prioritized. This could be achieved by choosing a laser repetition rate of 4 kHz (or lower) and setting a data frame of 37.5 km (or greater) to reduce the aerosol scattering ratio to

<1.10. Also, coincident passive measurements of stratospheric aerosol backscatter or extinction at a similar wavelength should be considered. Implementing these into a mission design would likely reduce the nighttime calibration uncertainties by a factor of two, which would then improve the accuracy of the resulting layer information and aerosol/cloud optical properties derived from the calibrated signal. Accurate backscatter lidar data is critical to improve our understanding of various physical properties of the atmosphere, specifically how clouds and aerosols radiatively impact our earth in the infrared.

## Acknowledgements

The CATS instrument was funded by the ISS NASA Research Office (NRO), with CATS data products and processing algorithms funded by NASA Science Mission Directorate (SMD). CATS data products are archived through the Atmospheric Science Data Center (ASDC). CATS browse images and data products are also freely distributed via the CATS web site at http://cats.gsfc.nasa.gov/data/. EARLINET acknowledges support from the European Union's Horizon 2020 research and innovation programme under grant agreements No 654109.

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

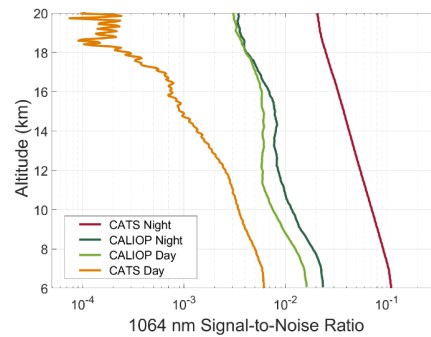


**Figure 1: The CATS and CALIOP 1064 nm signal to noise ratios for both daytime and nighttime data. The CATS nighttime SNR is nearly an order of magnitude greater than CALIOP (day and night), while the CATS daytime SNR is lower than CALIOP. The CATS profiles are computed for data acquired at a laser pulse rate of 4 kHz and averaged to 350 m horizontally. The CALIOP profiles are calculated for individual laser pulses acquired at 20.16 Hz, equivalent to a horizontal resolution of 335 m. The initial vertical resolution for all profiles is 60 m. All profiles are subsequently smoothed using a 2-km (34 point) running average.**

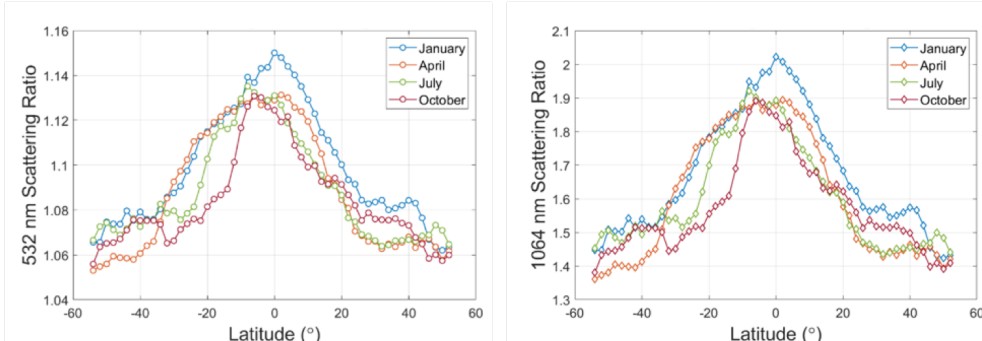

**Figure 2: The 532 nm scattering ratios measured by CALIOP within the CATS calibration region (left) and the 1064 nm scattering ratios estimated from the 532 nm retrievals (right) from 2016. These plots show the temporal and latitudinal variability within the calibration region where 1064 nm estimated scattering ratios can range from below 1.4 to above 2.0 depending on the time of year and geographical location.**

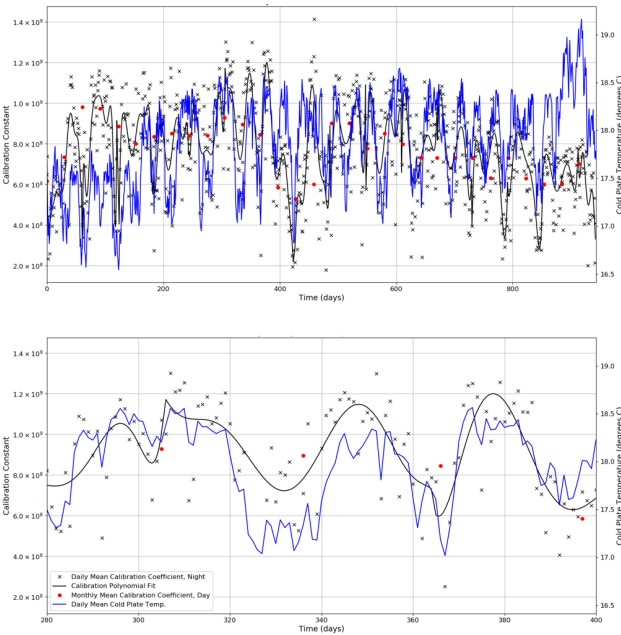

**Figure 3: The average CATS nighttime calibration coefficient for each day (black x), polynomial fit of the average calibration coefficient with time (black line), the CATS monthly daytime calibration coefficient (red circle), and the daily average cold plate temperature (blue line) for the entire mode 7.2 data record (April 2015 through October 2017) in the top panel. Zooming into a smaller time period (January 2016 through April 2016, bottom panel) demonstrates the correlation between calibration values and the instrument cold plate temperature. The correlation coefficient between the daily average nighttime calibration coefficient and cold plate temperature for the January-April 2016 period is 0.8066.**

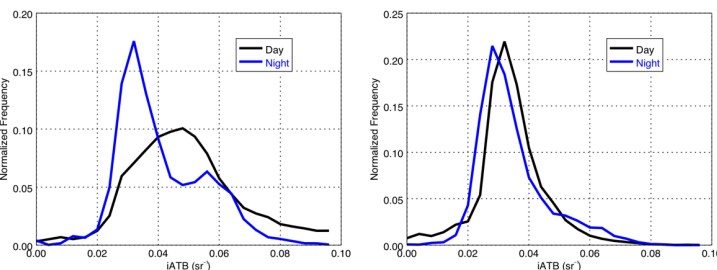

**Figure 4: Distributions of CATS strongly scattering, rapidly attenuating opaque cirrus iATB distributions from V2-01 (left) and V3-00 (right). These plots demonstrate the CATS daytime calibration method using calibration transfer from the nighttime calibration.**

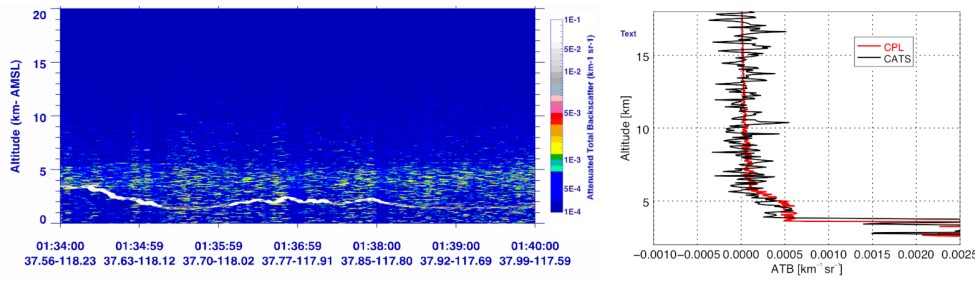

**Figure 5:  The CPL curtain plot of ATB centered around the 01:37 UTC coincident point from the 7-8 August 2015 CCAVE flight (Left). The mean ATB profiles from CATS and CPL during this under flight (right).**

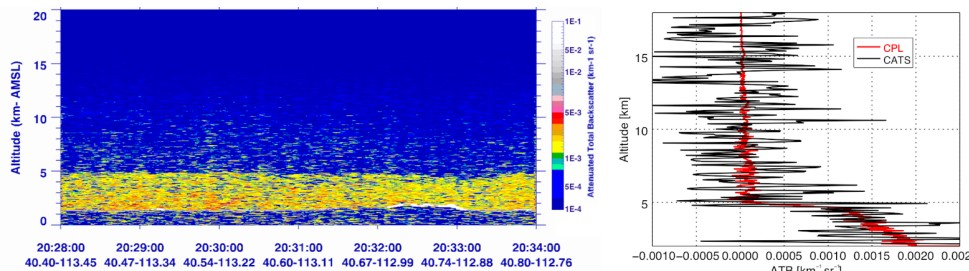

**Figure 6: The 20 August 2015 CATS/CPL coincident flight. The CPL 70 km coincident segment curtain plot (left) was used to compute the mean ATB profile (right) from CPL along the same path as CATS. The CATS and CPL data show good agreement despite higher noise levels in the CATS profile due to daytime retrieval limitations.**

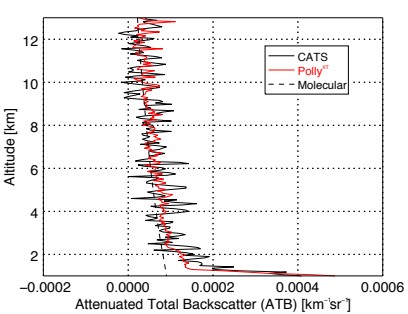

**Figure 7: The mean CATS and Polly$^{XT}$ ATB profiles from the CATS overpass of the Leipzig, Germany EARLINET site at 01:13:34 UTC on 24 September 2015. CATS passed within 31 km of the EARLINET site.**

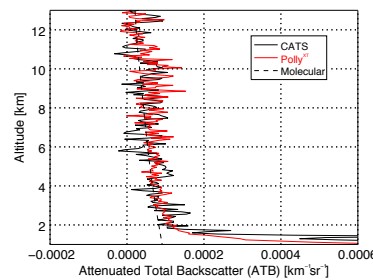

**Figure 8: The mean CATS and Polly$^{XT}$ ATB profiles from the CATS overpass of the Leipzig, Germany EARLINET site at 00:18:19 UTC on 30 July 2015. CATS passed within 41 km of the EARLINET site.**

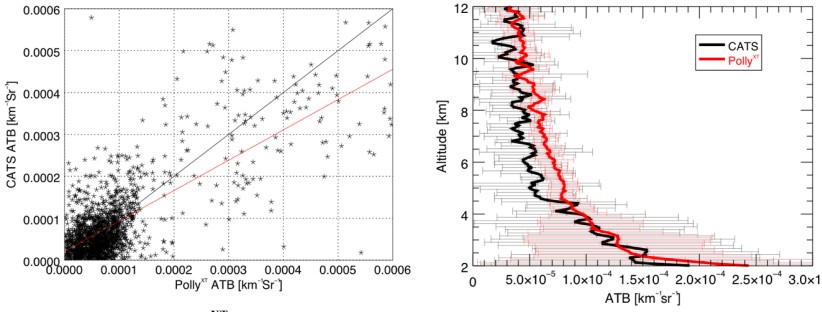

**Figure 9: Scatter plot of all eight Polly$^{XT}$ /CATS comparison overflights (left). The black line is the one-to-one line while the red line is the line fit of the data set. The correlation coefficient is 0.75. The average ATB profile from all eight Polly$^{XT}$/CATS comparison cases (right) shows the CATS mean profile is on average 19.67% lower than Polly$^{XT}$ from 3-12 km. The horizontal lines show the standard deviations of the mean profile for both CATS and Polly$^{XT}$.**

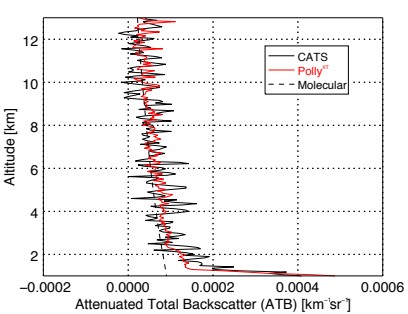

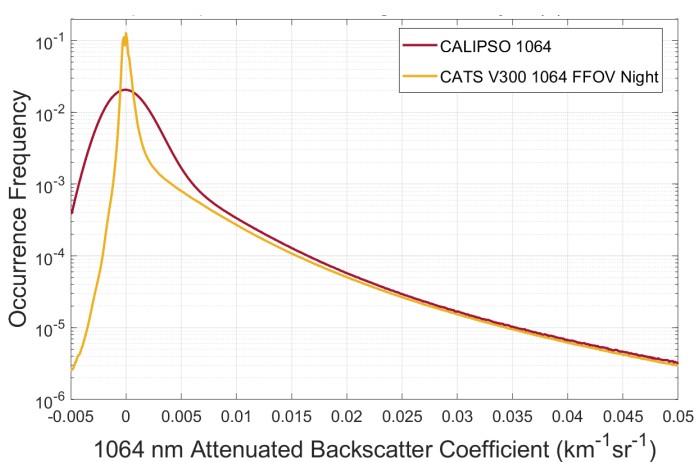

**Figure 10: Relative frequency distributions of 1064 nm attenuated backscatter coefficients measured by CALIOP (V4.10) and CATS (V3-00) from April through September 2016 at night with temperatures less than -40 C.**

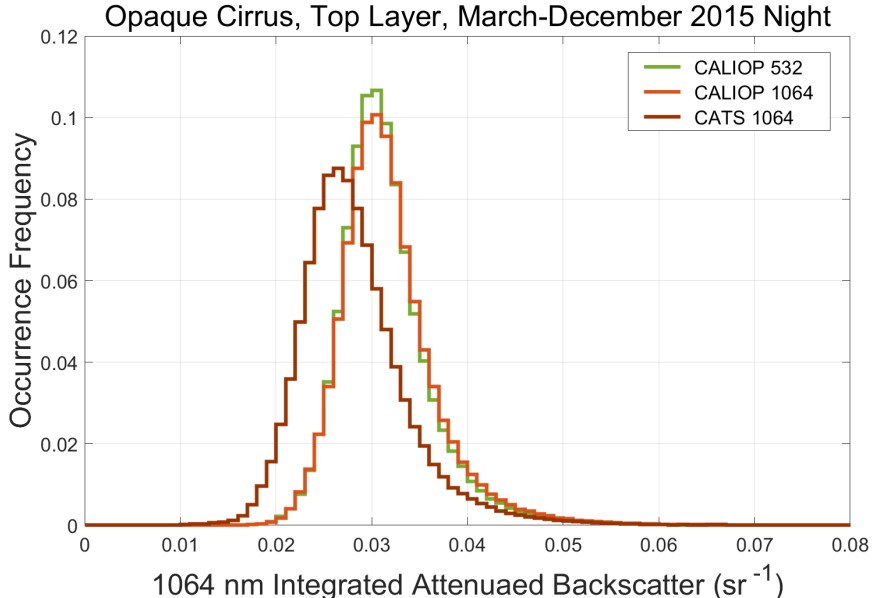

5    **Figure 11: Relative frequency distributions of March-December 2015 nighttime integrated attenuated backscatter for opaque cirrus clouds measured by CALIOP at 532 nm and 1064 nm and by CATS at 1064 nm only.**

|  | Night | Day | Mean Bias (Night-Day) | MAE |
|---|---|---|---|---|
| Mean | 0.03840 sr$^{-1}$ | 0.03823 sr$^{-1}$ | 0.000168 sr$^{-1}$ | 0.003419 sr$^{-1}$ |
| Median | 0.03559 sr$^{-1}$ | 0.03681 sr$^{-1}$ | -0.001215 sr$^{-1}$ | 0.003289 sr$^{-1}$ |

| Standard Dev. | 0.01386 sr$^{-1}$ | 0.01390 sr$^{-1}$ | -3.969e$^{-5}$ sr$^{-1}$ | 0.002430 sr$^{-1}$ |
|---|---|---|---|---|
| Mode | 0.02981 sr$^{-1}$ | 0.03239 sr$^{-1}$ | -0.00258 sr$^{-1}$ | 0.00413 sr$^{-1}$ |

**Table 1: Mean, median, mode and standard deviation of the day and night iATB distributions of rapidly attenuating, opaque cirrus clouds from all V3-00 CATS data. The mean bias, and mean absolute error (MAE) were also calculated between the day and night distributions.**

| | CALIOP 532 nm | CALIOP 1064 nm | CATS 1064 nm |
|---|---|---|---|
| minimum | 0.0017 sr$^{-1}$ | 0.0015 sr$^{-1}$ | 0.0001 sr$^{-1}$ |
| maximum | 0.1189 sr$^{-1}$ | 0.1248 sr$^{-1}$ | 0.1794 sr$^{-1}$ |
| median | 0.0303 sr$^{-1}$ | 0.0305 sr$^{-1}$ | 0.0270 sr$^{-1}$ |
| MAD | 0.0036 sr$^{-1}$ | 0.0038 sr$^{-1}$ | 0.0045 sr$^{-1}$ |
| mean | 0.0310 sr$^{-1}$ | 0.0313 sr$^{-1}$ | 0.0280 sr$^{-1}$ |
| standard deviation | 0.0050 sr$^{-1}$ | 0.0053 sr$^{-1}$ | 0.0071 sr$^{-1}$ |
| samples | 333,228 | 333,228 | 268,806 |

**Table 2: Descriptive statistics for the integrated attenuated backscatter of opaque cirrus clouds detected during nighttime granules by CATS and CALIOP during the period from 1 March to 31 December 2015 (MAD = median absolute distance).**