# Peer review of "Cloud Aerosol Transport System (CATS) 1064 nm Calibration and Validation"

_Atmospheric Measurement Techniques, 2019_

## Referee Comment (RC1) · Franco Marenco (Referee) · 21 Jun 2019

I have read the paper by Rebecca Pauly and co-authors with great interest. The article describes the calibration of CATS 1064 nm attenuated backscatter and depolarisation (level 1 data). Calibration is achieved on a per-granule basis, by normalisation of nighttime signals with modelled atmospheric backscatter at an altitude of 22-26 km, where account is made for Rayleigh scattering (derived from the MERRA-2 re-analysis) and aerosol scattering (inferred from CALIPSO measurements). Moreover, attenuated backscatter by opaque cirrus clouds is exploted for two further calibrations: daytime calibration, on a monthly basis, is achieved by matching the overall frequency distribution of daytime and nighttime opaque cirrus attenuated backscatter, and the calibration of depolarisation signals, on a yearly basis, is obtained by matching the parallel and

perpendicular signals for this type of clouds. The uncertainties that derive from this approach are discussed and quantified, and comparison with a number of validation sources is described: CALIPSO, airborne lidar, and ground-based lidar.

This research has a high significance, due to the fact that two and a half years of CATS data have been collected in 2015-2017, on-board the International Space Station. This dataset is still to be exploited in full, and it provides information on global aerosols and clouds, under an unusual orbit type (the one of the ISS) which permits an investigation on diurnal cycles (as opposed to the more traditional sun-synchronous orbits). It also demonstrates that, depending on instrument design, direct calibration of 1064 nm lidar channels is possible, without needing to transfer the calibration from channels at shorter wavelengths.

The paper is well written but a few more points need to be addressed before it can be published, in order to clarify better the methods to the reader. I feel that there are still a few major points as follows, some of which were already raised in the previous "quick review". Underpinning science to the CATS processing is addressed here and I believe that the explanation of the methodology should clarify all doubts.

MAJOR COMMENTS:

1) I suggest to add more in the conclusions. CATS has been used and will continue to be used for cloud, aerosol, and radiative budget studies that will benefit from the new data version. What are the most significant results obtained so far from CATS datasets? how would they be affected if they were to be reprocessed using V3 data? how does the V3 level 1 calibration affect the level 2 data (before any changes to the V3 level 2 processing)? are there any useful lessons from your research that can be useful for EarthCARE and Aeolus? and for future follow-on missions?

2) In the daytime calibration (section 2.2) , you specify that you are looking for a specific type of target: opaque and geometrically thin clouds, and you specify "A layer is classified as opaque if no layer or ground signal is detected below it". In the previous

review I raised the question of how you could know that such a cloud is physically thin, since opaque and deep clouds could look similar on a lidar signal. I don't believe that this point has been addressed.

3) Equation 9: discuss numerical differences between Cday and Cnight and their evolution; what causes them? instrument temperature?

4) Section 2, lines 6-26: A few pieces of information on the instrument, that one deducts whilst reading the paper, should go in this section, so that the reader can begin thinking about them. I would discuss the following in this section: (1) how laser 1 and laser 2 are associated with modes 7.1 and 7.2; (2) the difference in PRF between the two lasers (4 and 5 kHz); (3) the signal folding due to the choice of PRF; (4) how this is reflected in the signal acquisition (with a data frame from -2 to 28 km); (5) the raw vertical and temporal resolution; and (6) any integration that is applied to the data prior to the signal processing described in the present paper.

5) Equation 3: the colour ratio 0.4 is assumed because it is the value also assumed by Hair et al (2008). However, Hair et al do not give any explanation on why this value has been chosen, nor do they provide a reference! This should be discussed, and the error estimate on the colour ratio should be given explicitly. It could be useful to mention that this assumed colour ratio corresponds to a backscatter Angstrom exponent of 1.3, and that it is a colour ratio for "nearly clear air" (so is stated by Hair et al).

6) P5 L12-17: please explain these criteria better: (1) why has each of them been chosen and what do they signify in terms of cloud physics? (2) why do they differ from the criteria used for daytime calibration? (3) how is the temperature determined? (please state if it comes from the reanalysis); (4) clarify how you determine the depolarisation delta before you know PGR: are you using a previous data version for this? (5) equation 7 is missing the PGR as a multiplicative constant: have you already incorporated this into NRBperp? If this is the case, it is confusing, and I would recommend to write PGR explicitly, or to call NRBperp' = PGR * NRBperp.

7) is the daytime or nighttme calibration coefficient and the PGR stored in the level 1 data files available for download? please state in the paper.

8) The specifications of the units used is missing in several places: (1) Equation 1, what are the units for Ns? counts? count rate? voltage? and what are the units of NRB (e.g. counts * km2 / J)? (2) Equation 8, what are the units for C (e.g. counts * km2 * sr / j)? (3) P6 L7, specify the units with the calibration coefficients given here; (4) P7 L9, specify units of iATB values given (e.g. sr-1); Table 1 misses the specification of units (sr-1); etc.

MINOR COMMENTS:

9) The paper uses the normalisation technique to calibrate the signal; however, since aerosols have to be accounted for at the altitudes considered, I suggest that it should not be called a "molecular" normalisation technique. This can be achieved by removing the word "molecular" from lines 17 and 35 (abstract) and in a few places within the manuscript. In the conclusions, line 22, "Rayleigh profile" –> "Rayleigh profile corrected for aerosol contributions".

10) P2 L26: we have no measurements of crystal size, hence I would either remove the words "comprised of large ice crystals", or I would word it as a caveat (e.g. "thought to be mainly associated with ice crystals larger than the lidar wavelength").

11) P3 L16: How is the laser energy E determined? Is it measured on-board? Is E an instantaneous value, a nominal one, or an average over a given time period?

12) P3 L22: "averaging the signal acquired after the signal attenuated by the Earth's surface" add the words "after correction for the signal folding time (see below)".

13) P4 L6: You earlier specified that the data frame is limited to -2 to 28 km; the fact that you use signals between -2.5 and -4.5 km for the evaluation of the background seems in my opinion to contradict this fact. Please explain, and please specify whether the data frame between -2 and 28 is limited by hardware design (acquisition electronics).

14) P4 L23: "28 km" –> "26 km"

15) P5 L7: remove "reflected" (this is scattered light, rather than reflected).

16) P5 L22: add "(multiplied by PGR)" after "perpendicular"

17) P5 L24: add the following before "To prepare", so as to clarify to the reader better what is the overall approach: "Nighttime calibration is applied on a per-granule basis, where a single calibration coefficient is determined as follows, for each data granule".

18) P5 L33: specify the value used for minimum and maximum thresholds.

19) P6 L1-4: is there any flagging of cases where the per-granule approach fails and you revert to using the previous week data? or is it exactly coincident with the flagging of files with a poor depolarisation quality?

20) P6 L6-13: please explicitly state that an instrument temperature dependence is thought to be responsible for these fluctuations. Do you have any suggestion on which piece of the CATS hardware could be responsible?

21) P6 L22: precede line with "Instead,". "singular" –> "single". "month": specify if this is calendar month (from 1 to last day of the month) or a rolling 30-day period.

22) P6 L23: "colder than -20C": how is the temperature determined? see comment 6 on specifying how temperatures are determined.

23) Figure 4: why does the shape of the distribution change so much? I would only expect a horizontal shift on the plot.

24) P7 L6: add "on a monthly basis" after "V3-00".

25) P7 L8: I thouht that equation 9 would esnure that the bias on the mean would be zero. Please explain better why a residual bias persists.

26) P7 L28: "transmission" before "uncertainty".

27) P7 L37: please give a numerical estimate of (delta C / C)_sys before discussing

the random error.

28) P8 L11-12: if the multiple scattering factor is the same for daytime and nighttime measurements, does it not cancel out? please explain if it is different for day and night instead.

29) P9 L20: scattering ratio of 1.27: how much is the comparison with CATS sensitive to R? discuss the consequences of this assumption and its effect on the estimate of measurement errors; please specify if R is specified at 1054 or 532 nm.

30) P10 L2: "some of the differences in the ATB signal": I am not sure which differences you are referring to: the two do not look too different from each other!

31) P10 L30: add "PollyXT" before "1064 nm"

32) P10 L39: specify how many profiles are accumulated in 30 min of PollyXT measurements.

33) P10 L40: add "CATS-like" between "mean" and "signal"

34) P13 L30: please specify what changes to instrument design could permit the use of a higher calibration region. I suppose that one of them could be a reduction of the laser PRF (responsible for signal folding).

35) Figures 10 and 11, y-axis: please specify whether this is a relative frequency expressed in % or an absolute frequency distribution. In Figure 11 make the x-axis label consistent with Fig. 10 for a better readability.

---

## Referee Comment (RC2) · Anonymous Referee #4 · 2 Aug 2019

The paper describes an algorithm for calibrating and validation of CATS 1064nm backscatter coefficient. Overall, the work presented in this paper is very important because lidar observation at 1064 nm is needed together with 532 nm for characterizing particle size and other layered aerosol optical properties. The validation shows that the method appears to work well and gives an uncertainty of 20% when comparing with other lidar observations from different platforms.

I would recommend the paper be accepted after minor to moderate revisions to improve clarity and discuss its broader significance for the research community.

1) equations. the symbols in each equation should be well explained and with unit given (or otherwise mention unitless). This will help readers understand the equation better. For example, in equation 1, what is the unit of Ns, r, D, and E. In equation 2,

what is the unit of R, beta or backscatter coefficient. The list goes on for all equations.

2) equation 2. R is defined as aerosol scattering ratio. Should it be lidar ratio due to aerosol scattering? to separate it from aerosol single scattering albeit? How is it defined? Where does the equation (2) come from? If M is used to denote molecular, should A be added as a subscript for R because R is Aerosol scattering ratio? Again, description of unit and physics here will help to improve the clarity here.

3) paragraph before 2.2, what is the unit of calibration coefficient? what exactly is calibrated? from digital count to total attenuated backscatter coefficient? Table 2, the integrated attenuated backscatter has unit of sr-1? bur for CALIOP level-2 data, the same "total attenuated backscatter" has an unit of km-1sr-1. Given the terminologies can be used differently by different groups, it is important to define them from basic variables (e.g., extinction cross section, scattering phase function, etc) to avoid ambiguity.

4) conclusions. If the calibration has 20% uncertainty, does that also mean that the total aerosol optical depth derived from CATS will have an uncertainty of 20% at least? It is important to discuss the link between the calibration uncertainty and the level-2 product uncertainty.

5) finally, either in the introduction or conclusion, it is worthy to mention that lidar has been used to constrain smoke injection height (such as Wang et al., 2013, Atmospheric Research , 122, 486-503) and understand relative distrubiton of smoke and dust particles in the vertical (Yang et al., 2013, JGR, 118, 12,139-12,157) in the chemistry transport models.
* * *

---

## Short Comment (SC1) · 4 Aug 2019

**Comment on**
**"Cloud Aerosol Transport System (CATS) 1064 nm Calibration and Validation" by Pauly et al.**

Tyler J. Thorsen

NASA Langley Research Center; Hampton, VA

tyler.thorsen@nasa.gov

With several recent CATS science-related publications, this is a timely manuscript on the inner working of CATS and its retrievals. The wide-reach of CATS-related science applications certainly makes this manuscript appropriate for publication in AMT, eventually. However, in its current form, serious revisions are needed as there is a general lack of rigor and substantial ambiguity in the writing and analyses.

**Show the full variability of the coefficients**

The objective of this paper is to calculate calibration coefficients and their uncertainties, yet only a select 120 days of nighttime coefficients are shown in Fig. 3. I strongly encourage the authors to showcase the result of all their hard work: plot the entire 2.5+ years of calibration coefficients (night and day) along with their uncertainties. Just showing 120 nighttime coefficients over some unknown time period leaves several open questions in a reader's mind. How stable is the calibration? Are then trends/drifts in the calibrations and their uncertainties that a user should be aware of? Does the relationship with the cold plate temperature hold over the entire mission?

In addition to a time series of the coefficients/uncertainties, compositing these values by local time would be extremely useful information for the science community. The sampling throughout the diurnal cycle is one of CATS' most unique aspects that has, and will continue to, attract interest from those investigating diurnal cycles. However, any potential conclusions from these analyses needs to be tempered by the increased calibration uncertainty (and other uncertainties) during the daytime. Plotting and discussing the dependence of the calibration coefficients and their uncertainties with respect to local time would provide valuable context for those wishing to use CATS to study the diurnal cycle.

Showing how the calibration uncertainties vary is essential as they propagate into every aspect of the downstream science data products. The authors tend to focus on the effect of calibration uncertainties on the in-aerosol / in-cloud attenuated backscatter. But, before they are relevant there, they impact the detection thresholds. Somewhere, the authors should comment on how the calibration uncertainties and their diurnal variability impacts their feature detection. For example, presumably the increase in daytime calibration uncertainty necessitates more conservative detection thresholds to avoid any false positives. Have estimates been made of how many features may go undetected from this? This could be sussed out by imposing an artificial increase in the nighttime detection thresholds to see what features go undetected.

**Proper CALIOP comparisons**

The CATS and CALIOP comparisons in Section 2.1 and Fig. 1 need to be refined and expanded upon. First, the altitude range in Figure 1 should be extended to include both the CATS and CALIOP calibration range (i.e. up to 39km): the SNRs at these high altitudes are the relevant ones for calibration. In addition to comparing SNR in the entire column, as is done in Section 2.1, the text should be expanded to include a discussion of the SNR difference in each lidar's respective calibration regions. Additionally, since CALIOP performs its calibration at 532nm, the SNR profiles for CALIOP at 532nm should be added in as well and compared with CATS.

Despite the wavelength difference, comparing CATS 1064nm to CALIOP 532nm is more of an apples to apples comparison since they are the respective workhorse wavelengths for each lidar. These are the wavelengths for each lidar that are calibration, where feature detection is performed and the most accurate optical properties are available for. Therefore, comparing CATS 1064nm to

CALIOP 532nm is the most relevant comparison to those using the data products. However, as the authors do discuss, it is important to also point out CATS' superior SNR at 1064nm during nighttime for those whose particular investigations would benefit from this.

**Error analysis**

Section 3 is very unclear. Words like "overall" and "typical" are used to describe the various numbers and ranges given without any explanation of what they correspond to. Please be more precise when giving these numbers. Are these the average calibration uncertainties? Are the ranges interquartile ranges? Minimums and Maximums? Standard deviations? What is "var" in Eq. (13)? The authors seem to refer to this as "variability", do they mean variance? Please also indicate at what significance level the uncertainties presented here and in the data products are given for.

The uncertainty in the assumed backscatter color ratio does not appear to be included in the error analysis. Its value can vary substantially for (e.g. *Burton et al.*, 2012), which should be accounted for in the error budget. Alternatively, this could be avoided by just using the CALIOP 1064nm scattering ratios directly (see my comment below on this) and replacing the *Kar et al.* (2018) 532nm CALIOP calibration uncertainty with the 1064nm CALIOP calibration uncertainty given in *Vaughan et al.* (2019).

I was disappointed that the correlation between the nighttime calibration and the cold plate temperature was not exploited more (although it is not clear if relationship holds outside this 120 day period, see my comment above). I think the authors are missing out on an opportunity to explore improving their daytime calibration using this regression.

I would also suggest adding a short paragraph to the end of this Error Analysis section comparing the calibration uncertainties to other work that has performed a similar normalization, specifically MPLnet and CALIOP. Both of these where mentioned in the Introduction as forming the basis and background for this current study. Some brief context relative to MPLnet/CALIOP would make a nice connection back to your initial motivation and help give perspective to the readers that are more familiar with MPLnet/CALIOP than they are CATS.

**Validation**

I appreciated that the validation of this calibration tough: HSRL/Raman techniques aren't feasible at 1064nm, so all your left with is comparisons to other lidars who also need to calibration to a molecular signal. Because of this, one cannot treat CPL and EARLINET as absolute truth. Therefore, I suggest re-framing the discussion in section 4 around comparing the two profiles in the context of each instrument's calibration uncertainty (e.g. add uncertainty bars to the profiles Figs. 5-9 that correspond to each instrument's respective calibration uncertainty). That would put these comparisons within the proper context. Without this, it is easy to read too much into apparent absolute agreement as the authors themselves do on page 9 lines 29-30 where the agreement is called "surprising". The CPL/CATS agreement is NOT "surprising" after considering that the CPL was scaled by an assumed scattering ratio of 1.27! This large factor is quite uncertain and one could chose many reasonable values for it that would strongly impact the comparisons in Figs. 5 and 6. Showing the uncertainties involved would help avoid one reading too much into any agreement/disagreement.

From a sample size perspective, EARLINET is the authors' best bet for a comprehensive comparison. I encourage the authors' not to forgo this opportunity and go beyond only comparing eight nighttime overflights. I encourage the authors to also include daytime comparisons and a large

enough sample size to make meaningful statistical comparisons.

For the CALIOP comparison, what is the motivation for comparing attenuated backscatter in cirrus? Since this study is concerned with calibration, why not compare CALIOP and CATS attenuated backscatter profiles as was done in the CPL and EARLINET comparisons? Adding the complication of cloud detection and multiple scattering into this seems unnecessary and out of scope for this study.

**Aerosol scatting ratios**

The text describing the scattering ratios and their presentation in Fig. 2 is confusing and in need of revisions. First, as Reviewer 1 points out, it is not really fair to call this a "molecular" normalization technique since, as Fig 2. shows, aerosol comprises anywhere from 30–50% of the signal you're calibrating! This is a huge challenge/limitation that the authors aren't very up front about (see my comments in the next section concerning this). Considering the need for these scattering ratios and their large contribution to the overall uncertainty, the authors need to be more precise in describing how they are incorporated into the algorithm and build confidence that these values are accurate.

It is unclear how exactly these scattering ratios are applied. On page 4 the authors state that "the CALIOP data is used to estimate the spatially and temporally varying 1064 nm scattering ratio at these altitudes (Fig. 2)." In Fig. 2 zonal mean scattering ratios are plotted in 4 different months. Are these zonal means what is meant by "spatially and temporally"? Why are only 4 months plotted in Fig. 2? What scattering ratios are used for the months not plotted? More specifics are need here. Additionally, if zonal means are used, I would recommend putting standard deviations on the curves in Fig. 2 and discussing the amount of variability that is neglected by using mean values (this would need to be included in the error analysis as well).

For Eqs. (2) and (3): why not just use the CALIOP 1064nm scattering ratios directly instead of assuming a backscatter color ratio? As I mentioned above, the uncertainty in the assumed backscatter color ratio is likely larger than just using 1064nm CALIOP data directly. Plus, the error in 1064nm CALIOP scattering ratios has already been characterized (*Vaughan et al.*, 2019) which would make the authors' uncertainty analysis more straightforward.

Limb sounding instruments will, by far, give the highest accuracy scattering ratios at these altitudes. Did the authors explore any other alternatives to using CALIOP for getting the aerosol scattering ratios? At the very least, the CALIOP scattering ratios should be compared to the climatology of SAGE II, SAGE III, GOMOS, etc...

**Don't oversell the approach**

I've touched on this throughout my comments above. There are several statements throughout the paper that are misleading considering the large uncertainty in having a significant amount of aerosols in the calibration region and the reliance on CALIOP to account for this. The authors state in the abstract that "Overall, CATS has demonstrated that direct calibration of the 1064nm channel is possible from a space based lidar using the molecular normalization technique". But this statement is only half true because the CATS calibration relies on having another, already calibrated, lidar in space (CALIOP). You can't characterize this as a "direct calibration" if 30–50% of your calibration (i.e. Fig. 2) relies on inter-calibrating to CALIOP! In essence, the authors follow a similar approach as has been done in previous work: derive a 1064nm calibration from calibrated 532nm backscatter.

The are several instances of the authors being cagey about this. For example, on page 3 lines

30-31: "CATS exhibits high nighttime 1064 nm SNR, enabling 1064 nm attenuated total backscatter (ATB) direct calibration without any dependence on the CATS 532 nm signal". This statement is very misleading. Yes, you have no dependence on the CATS 532nm signal, but you do depend on the CALIOP 532nm signal and, on top of that, a CALIOP 532nm signal that is conveniently already calibrated for you. If the CATS 532nm signal was of sufficient quality, you would have certainty used the CATS 532nm signal instead!

Another example: page 13 lines 24-29. Here the authors do say the aerosol loading is higher in the CATS calibration region than for CALIOP. Instead of waiting for the conclusion, the authors should mention this in the introduction and then again when discussing Fig. 2. Additionally, it is important to convey, quantitatively, the difference between the two: CALIOP has aerosol scattering ratios that are less than 1.02 in its calibration regions (*Kar et al.*, 2018, their Figure 2b), CATS has values between 1.4-2.0 (Fig. 2). That is a very significant difference.

In spite of all these difficulties, the general pathway the authors have taken to calibration CATS is likely the best approach. But, the authors need to chose their words carefully and convey that their approach is not going to be widely applicable to other 1064nm lidars since they do not demonstrate an *independent* calibration of 1064nm backscatter. Speculating on a way to truly do so is good fodder for the conclusion. The authors do some of this already, but I would encourage them to expand that discussion a bit. Have the authors considered the precision/accuracy trade offs between a lowered rep rate and increasing the altitude of the calibration range? If the CATS measurements weren't limited to between 51S-51N how would polar stratospheric clouds impact the calibration in polar regions?

**More minor comments/edits**

The CATS 532nm channel is mentioned in the abstract and a few other places, but it is never explained why the Mode 7.2 532nm data isn't used. Why is its SNR so much lower? Does it not have the same photon counting detectors? Is the laser outputting less energy at 532nm? Is an attempt made to calibrate it at all? It is used for any data products?

I would suggest shortening the abstract. Some of the content is too specific for an abstract and difficult to understand without reading the main text first. Make sure the abstract acts as a stand-alone description of the paper.

page 1 line 15: "range-resolved", "vertical" and "profiles" are all redundant, choose one adjective here

page 2 line 6: change "Scientists have used various methods for calibrating" to "Various methods have been used for calibration". Also, this statement needs a citation(s): what are the various methods being referred to?

page 2 lines 9-11: change "Sometimes, as is the case for MPLNet," to "Since MPLNet". Also change "sites. In these cases, the aerosol optical depth" to ", the aerosol optical depth"

page 2 line 17: change "to differences in the" to "the weaker"

page 2 line 37: what wavelength

page 3 lines 37-38: any idea why this is? It is surprising a mere 1km would make such a difference.

page 5 lines 12-17: where did these criterion come from? The Liu et al (2004) study? If not, explain how they were chosen.

page 5 lines 18-22: these few sentences are confusing

page 5 line 33: give the thresholds used

page 5 line 37: does "file" = "granule"? These appear to be used interchangeably in a few other places as well.

page 5 line 37: "On average, 60-70%..." This is a range, not an average. Give the average value or explain what the range corresponds to.

page 7 line 26: don't give the total uncertainty until after discussing the individual contributions

page 7 lines 30-31 and Eq. (5): why not just use an updated model for molecular scattering?

page 6 line 17: replace "noise introduced by the solar background" to "solar background noise"

Figure 3: calibration spelled wrong in the caption

Section 3: make clear that you're deriving the nighttime uncertainty first in this section.

page 6 line 36: change "is shown" to "computed"

Figures 5 and 6: also show the CATS curtain in these figures

page 11 line 39: if this is a concern, why not just collocate them within some distance/time tolerance?

page 12 line 24: change "essentially" to "approximately"

page 12 line 5: remove "a detailed discussion of"

**References**

Burton, S. P., R. A. Ferrare, C. A. Hostetler, J. W. Hair, R. R. Rogers, M. D. Obland, C. F. Butler, A. L. Cook, D. B. Harper, and K. D. Froyd (2012), Aerosol classification using airborne High Spectral Resolution Lidar measurements – methodology and examples, *Atmos. Meas. Tech.*, *5*(1), 73–98, doi:10.5194/amt-5-73-2012.

Kar, J., M. A. Vaughan, K.-P. Lee, J. L. Tackett, M. A. Avery, A. Garnier, B. J. Getzewich, W. H. Hunt, D. Josset, Z. Liu, and et al. (2018), CALIPSO lidar calibration at 532 nm version 4 nighttime algorithm, *Atmos. Meas. Tech.*, *11*(3), 1459?1479, doi:10.5194/amt-11-1459-2018.

Vaughan, M., A. Garnier, D. Josset, M. Avery, K.-P. Lee, Z. Liu, W. Hunt, J. Pelon, Y. Hu, S. Burton, and et al. (2019), CALIPSO lidar calibration at 1064 nm version 4 algorithm, *Atmos. Meas. Tech.*, *12*(1), 51?82, doi:10.5194/amt-12-51-2019.

---

## Author Comment (AC1) · 12 Sep 2019

**Pauly et al. (2019), Response to Referees, 29 Aug. 2019**

Atmospheric Measurement Techniques Discussion
Response to Referees' Science Review Comments – August 2019

"Cloud-Aerosol Transport System (CATS) 1064 nm Calibration and Validation" - Pauly, R., J.E. Yorks, D.L. Hlavka, M.J. McGill, V. Amiridis, S.P. Palm, S.D. Rodier, M.A. Vaughan, P. Selmer, A.W. Kupchock, H. Baars, A. Gialitaki

We received referee comments from two referees and one document with short comments from a member of the scientific community. Our responses to the comments of two of the referees of our submission: amt-2019-172: Pauly et al., "Cloud-Aerosol Transport System (CATS) 1064 nm Calibration and Validation" are below. The referees were very helpful in clarifying our explanation of the method, as well as the importance to future missions and CATS retrievals. We hope the editor will find our responses address the major and minor comments of the referees. Our response to the short comments from the member of the scientific community will be provided in a separate document. We believe the manuscript is clearer and more robust, and we look forward to the new step towards publication. Note that the referee comments appear in black while our responses appear in red.

**Franco Morenco's Comments (Referee)**

I have read the paper by Rebecca Pauly and co-authors with great interest. The article describes the calibration of CATS 1064 nm attenuated backscatter and depolarization (level 1 data). Calibration is achieved on a per-granule basis, by normalization of nighttime signals with modelled atmospheric backscatter at an altitude of 22-26 km, where account is made for Rayleigh scattering (derived from the MERRA-2 re-analysis) and aerosol scattering (inferred from CALIPSO measurements). Moreover, attenuated backscatter by opaque cirrus clouds is exploted for two further calibrations: daytime calibration, on a monthly basis, is achieved by matching the overall frequency distribution of daytime and nighttime opaque cirrus attenuated backscatter, and the calibration of depolarisation signals, on a yearly basis, is obtained by matching the parallel and perpendicular signals for this type of clouds. The uncertainties that derive from this approach are discussed and quantified, and comparison with a number of validation sources is described: CALIPSO, airborne lidar, and ground-based lidar.

This research has a high significance, due to the fact that two and a half years of CATS data have been collected in 2015-2017, on-board the International Space Station. This dataset is still to be exploited in full, and it provides information on global aerosols and clouds, under an unusual orbit type (the one of the ISS) which permits an investigation on diurnal cycles (as opposed to the more traditional sun-synchronous orbits). It also demonstrates that, depending on instrument design, direct calibration of 1064 nm lidar channels is possible, without needing to transfer the calibration from channels at shorter wavelengths.

The paper is well written but a few more points need to be addressed before it can be published, in order to clarify better the methods to the reader. I feel that there are still a few major points as follows, some of which were already raised in the previous "quick review". Underpinning science to the CATS processing is addressed here and I believe that the explanation of the methodology should clarify all doubts.

MAJOR COMMENTS:

1) I suggest to add more in the conclusions. CATS has been used and will continue to be used for cloud, aerosol, and radiative budget studies that will benefit from the new data version. What are the most significant results obtained so far from CATS datasets? how would they be affected if they were to be reprocessed using V3 data? how does the V3 level 1 calibration affect the level 2 data (before any changes to the V3 level 2 processing)? are there any useful lessons from your research that can be useful for EarthCARE and Aeolus? and for future follow-on missions?

   a. More detailed discussion of the most significant results obtained from CATS data so far has been added to the first paragraph of the conclusion, in addition to the text added about how the calibration affects the L2 data products addressing Anonymous Referee #4's comment #4. Most studies using the retrievals of optical properties (e.g. extinction, optical depth) have used the V3-00 data. Since EarthCARE's lidar is a 355 nm HSRL and Aeolus is a 355 nm Doppler wind lidar, the 1064 nm atmospheric normalization technique shown here isn't very helpful for those missions. However, a sentence was added to the second paragraph of the conclusion to elaborate on how decreasing the laser repetition rate of a future CATS-like backscatter lidar could provide a larger data frame, and thus a higher calibration altitude (minor comment #34).

2) In the daytime calibration (section 2.2), you specify that you are looking for a specific type of target: opaque and geometrically thin clouds, and you specify "A layer is classified as opaque if no layer or ground signal is detected below it". In the previous review I raised the question of how you could know that such a cloud is physically thin, since opaque and deep clouds could look similar on a lidar signal. I don't believe that this point has been addressed.

   a. What we are really trying to say here is that for strongly scattering, rapidly attenuating opaque cirrus, there should be little difference between nighttime and daytime iATB retrievals. Thus, that is why we selected these types of clouds for the daytime calibration transfer procedure. We have added text in Section 2.2 (first and second paragraphs) to address this and have removed all mention of "physically thin" clouds from the paper and replaced it with the phrase "strongly scattering, rapidly attenuating opaque" clouds.

3) Equation 9: discuss numerical differences between Cday and Cnight and their evolution; what causes them? instrument temperature?

a. The time evolution of the CATS calibration coefficients is correlated to the thermal stability of the cooling loop on the ISS, which in turn is attributed to the changing of the sun's angle with respect to the ISS orbital plane, known as its beta angle. The CATS nighttime calibration coefficients oscillate from $4 \times 10^8$ to $1.4 \times 10^9$ km$^3$sr J$^{-1}$ counts with a period of roughly 30-40 days. This oscillation is a result of changes in the CATS laser properties (i.e. wavelength, alignment, energy) due to thermal instability of the cooling loop. The thermal instability of the cooling loop and instrument was monitored by the cold plate temperature. Text has been added to state all of these changes on page 6, lines 30-38. Also, Fig. 3 has been updated to include the entire mode 7.2 dataset (top, April 2015 – October 2017) as well as a subset from January- April 2016 (bottom). The daytime calibration coefficients for each month have been added as red dots. A discussion of the daytime calibration values, variability, and comparison to nighttime calibration coefficients has been added starting at pages 7, line 36. Unfortunately, the funding for producing CATS data products has expired. But, if we were to ever receive funding to create another version of the CATS data products, we would make more rapid estimates of the daytime calibration coefficients than the current monthly estimates.

4) Section 2, lines 6-26: A few pieces of information on the instrument, that one deducts whilst reading the paper, should go in this section, so that the reader can begin thinking about them. I would discuss the following in this section: (1) how laser 1 and laser 2 are associated with modes 7.1 and 7.2; (2) the difference in PRF between the two lasers (4 and 5 kHz); (3) the signal folding due to the choice of PRF; (4) how this is reflected in the signal acquisition (with a data frame from -2 to 28 km); (5) the raw vertical and temporal resolution; and (6) any integration that is applied to the data prior to the signal processing described in the present paper.

a. All of the requested information has been added to the first paragraph of Section 2.

5) Equation 3: the colour ratio 0.4 is assumed because it is the value also assumed by Hair et al (2008). However, Hair et al do not give any explanation on why this value has been chosen, nor do they provide a reference! This should be discussed, and the error estimate on the colour ratio should be given explicitly. It could be useful to mention that this assumed colour ratio corresponds to a backscatter Angstrom exponent of 1.3, and that it is a colour ratio for "nearly clear air" (so is stated by Hair et al).

a. To the authors' knowledge, the value or variability of the stratospheric aerosol backscatter color ratio is not documented in the literature. For the mean value, we follow Hair *et al.* (2008), so $\chi_P$ =0.40 is taken as a constant for the aerosol loading in the upper troposphere/ lower stratosphere. This value is originally derived from backscatter data shown in Spinhirne et al. (1997). Given that sulfate aerosols are potentially the largest contributor to the stratospheric aerosol loading, this value is also consistent with lower tropospheric measurements of sulfate aerosols. Text has been added to page 5, lines 6-9 that now states this. For the error estimate (variability), we performed an analysis of SAGE III extinction Angstrom exponent, averaged from June 2017 to August 2018 in the CATS calibration region, to find a

mean/standard deviation of 1.79 ± 0.10. We use this standard deviation as a relative uncertainty for the backscatter color ratio, so we assume an absolute uncertainty in the stratospheric aerosol backscatter color ratio of 0.024. This is now explained in the text on page 8, lines 26-33.

6) P5 L12-17: please explain these criteria better: (1) why has each of them been chosen and what do they signify in terms of cloud physics? (2) why do they differ from the criteria used for daytime calibration? (3) how is the temperature determined? (please state if it comes from the reanalysis); (4) clarify how you determine the depolarization delta before you know PGR: are you using a previous data version for this? (5) equation 7 is missing the PGR as a multiplicative constant: have you already incorporated this into NRBperp? If this is the case, it is confusing, and I would recommend to write PGR explicitly, or to call NRBperp' = PGR * NRBperp.

   a. These criteria are used to identify scenes with dense cirrus clouds that can be used to compute the PGR. These criteria are VERY similar to those used by CALIOP to identify cirrus clouds for their 1064 nm calibration transfer. We have now updated the text to clarify these things, as well as address 3, 4, and 5. These changes are on page 5, lines 28-34.

7) is the daytime or nighttime calibration coefficient and the PGR stored in the level 1 data files available for download? please state in the paper.

   a. Yes, both these values are stored in the Level 1B files, and the paper now states this on page 5, line 23 and page 6, line 28.

8) The specifications of the units used is missing in several places: (1) Equation 1, what are the units for Ns? counts? count rate? voltage? and what are the units of NRB (e.g. counts * km2 / J)? (2) Equation 8, what are the units for C (e.g. counts * km2 * sr / j)? (3) P6 L7, specify the units with the calibration coefficients given here; (4) P7 L9, specify units of iATB values given (e.g. sr-1); Table 1 misses the specification of units (sr-1); etc.

   a. Anonymous Referee #4 also made this point. Thank you for catching this detail. The appropriate units have been added to the latest version of the manuscript throughout. You can see examples in the text corresponding to Equation 1 (page 3) and Equation 5 (page 5).

MINOR COMMENTS:

9) The paper uses the normalisation technique to calibrate the signal; however, since aerosols have to be accounted for at the altitudes considered, I suggest that it should not be called a "molecular" normalisation technique. This can be achieved by removing the word "molecular" from lines 17 and 35 (abstract) and in a few places within the manuscript. In the conclusions, line 22, "Rayleigh profile" –> "Rayleigh profile corrected for aerosol contributions".

    a. Typically, this technique has always been called the Rayleigh or molecular normalization technique. However, most of the applications in the past were at 532 nm and in altitude regions with small aerosol contributions. Thus, we understand the referee's concerns. We have changed the phrases "Rayleigh/molecular normalization technique" to "atmospheric normalization technique" and "Rayleigh profile" to "Rayleigh profile corrected for aerosol contributions". A sentence on page 4, lines 16-19 defines this name.

10) P2 L26: we have no measurements of crystal size, hence I would either remove the words "comprised of large ice crystals", or I would word it as a caveat (e.g. "thought to be mainly associated with ice crystals larger than the lidar wavelength").

    a. Text was changed as suggested (page 2, line 30)

11) P3 L16: How is the laser energy E determined? Is it measured on-board? Is E an instantaneous value, a nominal one, or an average over a given time period?

    a. The laser energy per pulse is measured, then averaged onboard and reported at 20 Hz. Text was added to specify this (page 3, line 27)

12) P3 L22: "averaging the signal acquired after the signal attenuated by the Earth's surface" add the words "after correction for the signal folding time (see below)".

    a. Text was changed as suggested (page 3, line 35)

13) P4 L6: You earlier specified that the data frame is limited to -2 to 28 km; the fact that you use signals between -2.5 and -4.5 km for the evaluation of the background seems in my opinion to contradict this fact. Please explain, and please specify whether the data frame between -2 and 28 is limited by hardware design (acquisition electronics).

    a. This was a poorly chosen example. We have updated the text to use the example of -2.0 to 0.0 km, which includes signal from 37.5 to 39.5 km (page 4, line 24). Details of why the data frame was chosen to be -2 to 28 km are now provided in Section 2, first paragraph.

14) P4 L23: "28 km" –> "26 km"

    a. Text was changed as suggested (page 5, line 4)

15) P5 L7: remove "reflected" (this is scattered light, rather than reflected).

    a. The language was modified for accuracy.

16) P5 L22: add "(multiplied by PGR)" after "perpendicular"

    a. Text was changed as suggested (page 6, line 9)

17) P5 L24: add the following before "To prepare", so as to clarify to the reader better what is the overall approach: "Nighttime calibration is applied on a per-granule basis, where a single calibration coefficient is determined as follows, for each data granule".

    a. New text has been added based on these suggestions and comment #3 from Anonymous Referee #4.

18) P5 L33: specify the value used for minimum and maximum thresholds.

    a. These threshold values vary based on the fluctuations shown in Figure 3. We added text specifying this (page 6, line 22).

19) P6 L1-4: is there any flagging of cases where the per-granule approach fails and you revert to using the previous week data? or is it exactly coincident with the flagging of files with a poor depolarisation quality?

    a. Yes. The L1B data products include a Quality Control Flag. The 23rd bit of this flag denotes when historical calibration coefficients have been used. The text has been updated to state this (page 6, line 30). This was a very helpful comment.

20) P6 L6-13: please explicitly state that an instrument temperature dependence is thought to be responsible for these fluctuations. Do you have any suggestion on which piece of the CATS hardware could be responsible?

    a. Please see our response to comments #3. We believe that the laser, given it is the most impacted by cooling loop temperatures, is leading to the changes in the calibration coefficients, but we don't have enough engineering data to determine what property of the laser is the source (wavelength shift, alignment with telescope, etc.).

21) P6 L22: precede line with "Instead,". "singular" –> "single". "month": specify if this is calendar month (from 1 to last day of the month) or a rolling 30-day period.

    a. This sentence has been modified based on the referee's comments (page 7, line 12).

22) P6 L23: "colder than -20C": how is the temperature determined? see comment 6 on specifying how temperatures are determined.

    a. We added text specifying that the temperature comes from MERRA-2 data (page 7, line 14).

23) Figure 4: why does the shape of the distribution change so much? I would only expect a horizontal shift on the plot.

    a. As stated in lines 37-38 of page 7, the changes in distribution are attributed to changes in the layer typing algorithms implemented in CATS V3-00 data. For example, the V3-00 cloud phase algorithm removed the secondary peak in the nighttime distribution at 0.055 sr-1 that was due to misclassification of liquid water clouds. See the CATS ATBD for more information about the feature identification algorithms. Yorks et al. (in prep) will include an update on these algorithms for V3-00.

24) P7 L6: add "on a monthly basis" after "V3-00".

    a. Text was changed as suggested (page 7, line 37)

25) P7 L8: I thought that equation 9 would ensure that the bias on the mean would be zero. Please explain better why a residual bias persists.

a. The bias between the daytime and nighttime iATB is introduced by the temporal resolution of the calibration coefficient at nighttime (1 per file) and daytime (1 per month). This is now stated on page 8, lines 1-4.

26) P7 L28: "transmission" before "uncertainty".

a. Text was changed as suggested (page 8, line 23)

27) P7 L37: please give a numerical estimate of (delta C / C)_sys before discussing the random error.

a. The system error is estimated as 7%, and is now reported in the paper on page 8, line 35.

28) P8 L11-12: if the multiple scattering factor is the same for daytime and nighttime measurements, does it not cancel out? please explain if it is different for day and night instead.

a. Yes, you are correct that the multiple scattering factor would cancel out. That entire paragraph in Section 3 has been rewritten.

29) P9 L20: scattering ratio of 1.27: how much is the comparison with CATS sensitive to R? discuss the consequences of this assumption and its effect on the estimate of measurement errors; please specify if R is specified at 1054 or 532 nm.

a. We now specify that the scattering ratio is the 1064 nm particulate scattering ratio (page 10, line 27), and we recognize that this scattering ratio could be a reason for the better than expected agreement between CPL and CATS for this case (line 39, page 10).

30) P10 L2: "some of the differences in the ATB signal": I am not sure which differences you are referring to: the two do not look too different from each other!

a. We have deleted this sentence since it doesn't really add value to the discuss of the comparison of CPL and CATS for the 7-15 km altitude region.

31) P10 L30: add "PollyXT" before "1064 nm"

a. Text was added as suggested (page 11, line 39)

32) P10 L39: specify how many profiles are accumulated in 30 min of PollyXT measurements.

a. PollyXT has a repetition rate of 20 Hz and they accumulate 30 seconds (i.e. 600 laser shots or single profiles). For a 30-min measurement segment, this makes 60*600=36000 single profiles. This is now specified on page 12, line 8.

33) P10 L40: add "CATS-like" between "mean" and "signal"

a. Text was added as suggested (page 12, line 10)

34) P13 L30: please specify what changes to instrument design could permit the use of a higher calibration region. I suppose that one of them could be a reduction of the laser PRF (responsible for signal folding).

a. As discussed in our response to comment #1, a sentence was added to the second paragraph of the conclusion to elaborate on how decreasing the laser repetition rate of a future CATS-like backscatter lidar could provide a larger data frame, and thus a higher calibration altitude.

35) Figures 10 and 11, y-axis: please specify whether this is a relative frequency expressed in % or an absolute frequency distribution. In Figure 11 make the x-axis label consistent with Fig. 10 for a better readability.

a. We have updated the text to specify that this is a relative frequency and remade Figure 11 with an updated x-axis label.

---

## Author Comment (AC2) · 12 Sep 2019

Atmospheric Measurement Techniques Discussion
Response to Referees' Science Review Comments – August 2019

"Cloud-Aerosol Transport System (CATS) 1064 nm Calibration and Validation" - Pauly, R., J.E. Yorks, D.L. Hlavka, M.J. McGill, V. Amiridis, S.P. Palm, S.D. Rodier, M.A. Vaughan, P. Selmer, A.W. Kupchock, H. Baars, A. Gialitaki

We received referee comments from two referees and one document with short comments from a member of the scientific community. Our responses to the comments of two of the referees of our submission: amt-2019-172: Pauly et al., "Cloud-Aerosol Transport System (CATS) 1064 nm Calibration and Validation" are below. The referees were very helpful in clarifying our explanation of the method, as well as the importance to future missions and CATS retrievals. We hope the editor will find our responses address the major and minor comments of the referees. Our response to the short comments from the member of the scientific community will be provided in a separate document. We believe the manuscript is clearer and more robust, and we look forward to the new step towards publication. Note that the referee comments appear in black while our responses appear in red.

**Anonymous Referee #4 Comments**

The paper describes an algorithm for calibrating and validation of CATS 1064nm backscatter coefficient. Overall, the work presented in this paper is very important because lidar observation at 1064 nm is needed together with 532 nm for characterizing particle size and other layered aerosol optical properties. The validation shows that the method appears to work well and gives an uncertainty of 20% when comparing
with other lidar observations from different platforms. I would recommend the paper be accepted after minor to moderate revisions to improve clarity and discuss its broader significance for the research community.

1) equations. the symbols in each equation should be well explained and with unit given (or otherwise mention unitless). This will help readers understand the equation better. For example, in equation 1, what is the unit of Ns, r, D, and E. In equation 2, what is the unit of R, beta or backscatter coefficient. The list goes on for all equations.
    a. The appropriate units have been added to the latest version of the manuscript throughout. You can see examples in the text corresponding to Equation 1 (page 3) and Equation 5 (page 5). Thank you for pointing out where they were missing.

2) equation 2. R is defined as aerosol scattering ratio. Should it be lidar ratio due to aerosol scattering? to separate it from aerosol single scattering albeit? How is it defined? Where does the equation (2) come from? If M is used to denote molecular, should A be added as a subscript for R because R is Aerosol scattering ratio? Again, description of unit and physics here will help to improve the clarity here.

    a. We have re-worked this section for clarity. We now provide a generic definition for the particulate scattering ratio (lines 31-34, page 4), which should prevent confusion about lidar ratio vs. particulate scattering ratio. We have also swapped Equations 2 and 3 so that it is more obvious how they relate. Finally, we more clearly label the variables as M to denote molecular and P to denote particulate.

3) paragraph before 2.2, what is the unit of calibration coefficient? what exactly is calibrated? from digital count to total attenuated backscatter coefficient? Table 2, the integrated attenuated backscatter has unit of sr-1? bur for CALIOP level-2 data, the same "total attenuated backscatter" has an unit of km-1sr-1. Given the terminologies can be used differently by different groups, it is important to define them from basic variables (e.g., extinction cross section, scattering phase function, etc) to avoid ambiguity.

    a. The units of the calibration coefficient are now provided on page 6, line 14 (and throughout the paper). More details about what this calibration coefficient is being applied to (the NRB profile) and the result (the ATB profile) are provided on lines 8-9 on page 6. The units of total attenuated backscatter (or attenuated total backscatter), which are $km^{-1}sr^{-1}$, are different than the integrated attenuated backscatter ($sr^{-1}$). In the equation for integrated attenuated backscatter,

$$\gamma' = \int_{top}^{base} \beta'(r)\, dr,$$ the differential range element dr has units of km, so the

integrated quantity, $\gamma'$, has units of $sr^{-1}$. These units are the same for both CATS and CALIOP.

4) conclusions. If the calibration has 20% uncertainty, does that also mean that the total aerosol optical depth derived from CATS will have an uncertainty of 20% at least? It is important to discuss the link between the calibration uncertainty and the level-2 product uncertainty.

    a. In general, yes, a calibration uncertainty of ±10-20% imposes a lower bound of ±10-20% on the uncertainty of the optical depth retrievals. We have added some brief text to the conclusion to express this relationship. However, the propagation of calibration errors in the solution of the lidar equation is both nonlinear and non-trivial, hence a more complete discussion of the link between calibration uncertainty and level 2 product uncertainties lies well beyond the scope of this paper. A complete mathematical description of calibration error propagation for elastic backscatter lidar measurements is given by Young et al., 2013 and Young et al., 2016.

5) finally, either in the introduction or conclusion, it is worthy to mention that lidar has been used to constrain smoke injection height (such asWang et al., 2013, Atmospheric Research , 122, 486-503) and understand relative distrubiton of smoke and dust particles

in the vertical (Yang et al., 2013, JGR, 118, 12,139-12,157) in the chemistry transport models.

   a.   References and discussion of these lidar applications were added to the introduction on page 2, lines 1-2.

---

## Author Comment (AC3) · 12 Sep 2019

Atmospheric Measurement Techniques Discussion
Response to Short Comments by Tyler Thorsen – September 2019

"Cloud-Aerosol Transport System (CATS) 1064 nm Calibration and Validation" - Pauly, R., J.E. Yorks, D.L. Hlavka, M.J. McGill, V. Amiridis, S.P. Palm, S.D. Rodier, M.A. Vaughan, P. Selmer, A.W. Kupchock, H. Baars, A. Gialitaki

We received short comments from Tyler Thorsen (tyler.thorsen@nasa.gov) of NASA Langley Research Center. Some of his comments are also raised by the two referees of the manuscript. Where appropriate, we will echo those responses here in this document. Our responses appear below in red.
* * *
Comment on "Cloud Aerosol Transport System (CATS) 1064 nm Calibration and Validation" by Pauly et al.

Tyler J. Thorsen
NASA Langley Research Center; Hampton, VA
tyler.thorsen@nasa.gov

With several recent CATS science-related publications, this is a timely manuscript on the inner working of CATS and its retrievals. The wide-reach of CATS-related science applications certainly makes this manuscript appropriate for publication in AMT, eventually. However, in its current form, serious revisions are needed as there is a general lack of rigor and substantial ambiguity in the writing and analyses.

**Show the full variability of the coefficients**
The objective of this paper is to calculate calibration coefficients and their uncertainties, yet only a select 120 days of nighttime coefficients are shown in Fig. 3. I strongly encourage the authors to showcase the result of all their hard work: plot the entire 2.5+ years of calibration coefficients (night and day) along with their uncertainties. Just showing 120 nighttime coefficients over some unknown time period leaves several open questions in a reader's mind. How stable is the calibration? Are then trends/drifts in the calibrations and their uncertainties that a user should be aware of? Does the relationship with the cold plate temperature hold over the entire mission?

One of our referees also asked us to expand on the evolution of the CATS nighttime and daytime calibration coefficients and what specifically causes the fluctuations. We echo our response here.

The time evolution of the CATS calibration coefficients is correlated to the thermal stability of the cooling loop on the ISS, which in turn is attributed to the changing of the sun's angle with respect to the ISS orbital plane, known as its beta angle. The CATS nighttime calibration coefficients oscillate from $4\times10^8$ to $1.4\times10^9$ km$^3$sr J$^{-1}$ counts with a period of roughly 30-40 days. This

oscillation is a result of changes in the CATS laser properties (i.e. wavelength, alignment, energy) due to thermal instability of the cooling loop. The thermal instability of the cooling loop and instrument was monitored by the cold plate temperature. Text has been added to state all of these changes on page 6, lines 30-38. Also, Fig. 3 has been updated to include the entire mode 7.2 dataset (top, April 2015 – October 2017) as well as a subset from January- April 2016 (bottom). The daytime calibration coefficients for each month have been added as red dots. A discussion of the daytime calibration values, variability, and comparison to nighttime calibration coefficients has been added starting at pages 7, line 36. Unfortunately, the funding for producing CATS data products has expired. But, if we were to ever receive funding to create another version of the CATS data products, we would make more rapid estimates of the daytime calibration coefficients than the current monthly estimates.

In addition to a time series of the coefficients/uncertainties, compositing these values by local time would be extremely useful information for the science community. The sampling throughout the diurnal cycle is one of CATS' most unique aspects that has, and will continue to, attract interest from those investigating diurnal cycles. However, any potential conclusions from these analyses needs to be tempered by the increased calibration uncertainty (and other uncertainties) during the daytime.  Plotting and discussing the dependence of the calibration coefficients and their uncertainties with respect to local time would provide valuable context for those wishing to use CATS to study the diurnal cycle. Showing how the calibration uncertainties vary is essential as they propagate into every aspect of the downstream science data products. The authors tend to focus on the effect of calibration uncertainties on the in-aerosol / in-cloud attenuated backscatter. But, before they are relevant there, they impact the detection thresholds. Somewhere, the authors should comment on how the calibration uncertainties and their diurnal variability impacts their feature detection. For example, presumably the increase in daytime calibration uncertainty necessitates more conservative detection thresholds to avoid any false positives. Have estimates been made of how many features may go undetected from this? This could be sussed out by imposing an artificial increase in the nighttime detection thresholds to see what features go undetected.

Since the CATS daytime calibration coefficients are computed on a monthly basis and the CATS nighttime calibration coefficients oscillate with a period of roughly 30-40 days, there is little change, if any (daytime), in the calibration values throughout the diurnal cycle on the Earth's surface for a given day. There are day/night differences, and these can be seen in Figure 3. We agree that calibration uncertainties propagate into every aspect of the downstream science data products. This is undoubtedly an interesting topic, but beyond the scope of this paper (it is a paper in itself). Based on a referee comment, we have added some brief text to the conclusion to express the relationship between calibration and data products such as optical depth and extinction.

**Proper CALIOP comparisons**
The CATS and CALIOP comparisons in Section 2.1 and Fig. 1 need to be refined and expanded upon. First, the altitude range in Figure 1 should be extended to include both the CATS and

CALIOP calibration range (i.e. up to 39km): the SNRs at these high altitudes are the relevant ones for calibration.

We did not address this comment in the latest version of the paper, for the following reasons:

a)  Since CALIOP does not use the molecular normalization technique to calibrate its 1064 nm channel, computing molecular backscatter SNR at high altitudes still will not yield an apples-to-apples comparison. The reason that CALIOP does not use the molecular normalization technique at 1064 nm is that its high-altitude molecular SNR is far too poor. The intent of Figure 1 is to demonstrate that the CATS nighttime SNR is substantially larger than CALIOP's, and hence the atmospheric normalization technique is indeed viable *at night*.

b)  CALIOP does not make 1064 nm measurements above 30 km.

c)  The 8.2-to-20.2 km comparison region was chosen to illustrate SNR differences simply because the vertical and horizontal resolutions of the CATS and CALIOP level 1 data products are almost identical there (~60 m vertical for both instruments; ~350 m horizontally for CATS vs. ~335 m for CALIOP).

In addition to comparing SNR in the entire column, as is done in Section 2.1, the text should be expanded to include a discussion of the SNR difference in each lidar's respective calibration regions. Additionally, since CALIOP performs its calibration at 532 nm, the SNR profiles for CALIOP at 532 nm should be added in as well and compared with CATS.

On this point we disagree.  What's relevant to this paper is assessing the accuracy of the CATS 1064 nm calibration. The 532 nm SNR in the CALIOP calibration regions is relevant to this exercise only insofar as it contributes to the error budget of the CALIOP 1064 nm calibration coefficients (e.g., see section 4.3 in Vaughan et al., 2019).

Despite the wavelength difference, comparing CATS 1064 nm to CALIOP 532 nm is more of an apples to apples comparison since they are the respective workhorse wavelengths for each lidar. These are the wavelengths for each lidar that are calibration, where feature detection is performed and the most accurate optical properties are available for. Therefore, comparing CATS 1064 nm to CALIOP 532 nm is the most relevant comparison to those using the data products.

To repeat a response to a referee comment, "the propagation of calibration errors in the solution of the lidar equation is both nonlinear and non-trivial, hence a more complete discussion of the link between calibration uncertainty and Level 2 data product uncertainties lies well beyond the scope of this paper.  A complete mathematical description of calibration error propagation for elastic backscatter lidar measurements is given by Young et al., 2013 and Young et al., 2016." We note that the development given in the Young papers is wavelength agnostic, and thus would apply in equal measure to CALIOP at 532 nm and CATS at 1064 nm.

However, as the authors do discuss, it is important to also point out CATS' superior SNR at 1064 nm during nighttime for those whose particular investigations would benefit from this.

**Error analysis**
Section 3 is very unclear. Words like "overall" and "typical" are used to describe the various numbers and ranges given without any explanation of what they correspond to. Please be more precise when giving these numbers. Are these the average calibration uncertainties? Are the ranges interquartile ranges? Minimums and Maximums? Standard deviations? What is "var" in Eq. (13)? The authors seem to refer to this as "variability", do they mean variance? Please also indicate at what significance level the uncertainties presented here and in the data products are given for.

The words "overall' and "typical" have been removed from Section 3 and Equation 13 has been updated as suggested.

The uncertainty in the assumed backscatter color ratio does not appear to be included in the error analysis.

Equation 10 has been updated and now includes a color ratio term.  Regarding the evaluation of this term we will repeat a response to one of the manuscript's referees.

To the authors' knowledge, the value or variability of the stratospheric aerosol backscatter color ratio is not documented in the literature. For the mean value, we follow Hair *et al.* (2008), so $\chi_P$ =0.40 is taken as a constant for the aerosol loading in the upper troposphere/ lower stratosphere. This value is originally derived from backscatter data shown in Spinhirne et al. (1997). Given that sulfate aerosols are potentially the largest contributor to the stratospheric aerosol loading, this value is also consistent with lower tropospheric measurements of sulfate aerosols. Text has been added to page 5, lines 6-9 that now states this.

Its value can vary substantially for (e.g. Burton et al., 2012), which should be accounted for in the error budget.

We disagree with this statement. For aerosols in the lower troposphere, as discussed in Burton et al. (2012), backscatter color ratios can vary substantially because of shorter lifetimes and plumes that are more heterogenous. For aerosols in the stratosphere, as discussed here in our AMTD paper, backscatter color ratios are not expected to vary much (unless a large volcanic eruption occurs that injects aerosols into the stratosphere – and no eruptions injected aerosols into the CATS calibration altitude region during the 33 months of operation) because aerosols in the stratosphere have longer lifetimes and plumes are more homogenous. A nice general overview of stratospheric aerosol lifetimes and properties is given by Kremser et al. (2016). To estimate error (or variability) in the backscatter color ratio, we performed an analysis of SAGE III extinction Angstrom exponent, averaged from June 2017 to August 2018 in the CATS calibration region, to find a mean/standard deviation of $1.79 \pm 0.10$. We use this standard deviation as a relative uncertainty for the backscatter color ratio (6%), so we assume a stratospheric aerosol backscatter color ratio of $0.400 \pm 0.024$. This is now explained in the text on page 8, lines 26-33.

Kremser, S., et al. (2016), Stratospheric aerosol—Observations, processes, and impact on climate, Rev. Geophys.,54,278–335, doi:10.1002/2015RG000511.

Alternatively, this could be avoided by just using the CALIOP 1064 nm scattering ratios directly (see my comment below on this) and replacing the Kar et al. (2018) 532 nm CALIOP calibration uncertainty with the 1064 nm CALIOP calibration uncertainty given in Vaughan et al. (2019).

Unfortunately, it's not that straightforward. The reason that we did not use the CALIOP 1064 nm scattering ratios is the same reason CALIOP does not use the molecular normalization technique at 1064 nm - its high-altitude molecular SNR is far too poor.

I was disappointed that the correlation between the nighttime calibration and the cold plate temperature was not exploited more (although it is not clear if relationship holds outside this 120 day period, see my comment above). I think the authors are missing out on an opportunity to explore improving their daytime calibration using this regression.

See our previous comments on the relationship between the calibration and the cold plate temperature.

I would also suggest adding a short paragraph to the end of this Error Analysis section comparing the calibration uncertainties to other work that has performed a similar normalization, specifically MPLnet and CALIOP. Both of these where mentioned in the Introduction as forming the basis and background for this current study. Some brief context relative to MPLnet/CALIOP would make a nice connection back to your initial motivation and help give perspective to the readers that are more familiar with MPLnet/CALIOP than they are CATS.

This information is well documented in the literature and referenced in this paper.

**Validation**
I appreciated that the validation of this calibration tough: HSRL/Raman techniques aren't feasible at 1064 nm, so all your left with is comparisons to other lidars who also need to calibration to a molecular signal. Because of this, one cannot treat CPL and EARLINET as absolute truth. Therefore, I suggest re-framing the discussion in section 4 around comparing the two profiles in the context of each instrument's calibration uncertainty (e.g. add uncertainty bars to the profiles Figs. 5-9 that correspond to each instrument's respective calibration uncertainty). That would put these comparisons within the proper context. Without this, it is easy to read too much into apparent absolute agreement as the authors themselves do on page 9 lines 29-30 where the agreement is called "surprising".

In some instances, we have already done this. For example, in Section 4.2 we say "The difference between the two instruments falls within the uncertainties in the CATS ATB (Sect. 3) and the uncertainties in the Polly[XT] retrievals." We have edited parts of Sections 4.1 and 4.3 to re-frame the discussion as suggested.

The CPL/CATS agreement is NOT "surprising" after considering that the CPL was scaled by an assumed scattering ratio of 1.27! This large factor is quite uncertain and one could chose many

reasonable values for it that would strongly impact the comparisons in Figs. 5 and 6. Showing the uncertainties involved would help avoid one reading too much into any agreement/disagreement.

Our "assumed scattering ratio of 1.27" was not hand-picked out of thin air simply to get better agreement between CATS and CPL. Instead, this value comes straight from Table 4 of Vaughan et al. (2010), as discussed on page 10, line 28 of the manuscript. While we saw good agreement (2.28%) in one case, we also found a difference of 20.88% in the other case.

From a sample size perspective, EARLINET is the authors' best bet for a comprehensive comparison. I encourage the authors' not to forgo this opportunity and go beyond only comparing eight nighttime overflights. I encourage the authors to also include daytime comparisons and a large enough sample size to make meaningful statistical comparisons.

The reason for limiting the EARLINET comparisons to these 8 nighttime cases was twofold. (1) The only aircraft CATS underflight data we have are 2 daytime cases with CPL, so we wanted to show some nighttime comparisons to an independent lidar measurement. (2) These EARLINET cases represent times that CATS was in close proximity to the ground sites and the ground lidars were operational. For more comparisons between CATS and EARLINET, we suggest checking out a new paper published in ACPD below.

Proestakis, E., et al. (2019), EARLINET evaluation of the CATS L2 aerosol backscatter coefficient product, *Atmos. Chem. Phys. Discuss.*, https://doi.org/10.5194/acp-2019-45.

For the CALIOP comparison, what is the motivation for comparing attenuated backscatter in cirrus? Since this study is concerned with calibration, why not compare CALIOP and CATS attenuated backscatter profiles as was done in the CPL and EARLINET comparisons? Adding the complication of cloud detection and multiple scattering into this seems unnecessary and out of scope for this study.

Yes, ideally, we would compare heavily averaged attenuated backscatter profiles in some high-altitude region where the aerosol loading is both low and temporally stable. Unfortunately, as clearly indicated by Figure 1 in the ATMD manuscript, the CALIOP 1064 nm backscatter signal is too noisy to support reliable comparisons of backscatter signals having very low aerosol scattering ratios. This low SNR is a consequence of the CALIOP 1064 nm detectors (avalanche photodiodes), which are plagued with a huge amount of dark noise. The magnitude of this dark noise is clearly evident by examining the CALIOP 1064 nm signal distributions around $\beta'_{1064} = 0$ in Figure 10 of the ATMD manuscript.

**Aerosol scatting ratios**
The text describing the scattering ratios and their presentation in Fig. 2 is confusing and in need of revisions. First, as Reviewer 1 points out, it is not really fair to call this a "molecular" normalization technique since, as Fig 2. shows, aerosol comprises anywhere from 30–50% of the signal you're calibrating! This is a huge challenge/limitation that the authors aren't very up front about (see my comments in the next section concerning this). Considering the need for these

scattering ratios and their large contribution to the overall uncertainty, the authors need to be more precise in describing how they are incorporated into the algorithm and build confidence that these values are accurate.

We will address this comment in the next section.

It is unclear how exactly these scattering ratios are applied. On page 4 the authors state that "the CALIOP data is used to estimate the spatially and temporally varying 1064 nm scattering ratio at these altitudes (Fig. 2)." In Fig. 2 zonal mean scattering ratios are plotted in 4 different months. Are these zonal means

yes

what is meant by \spatially and temporally"?

Zonal means are inherently spatial averages. We describe the temporal averaging on page 5, lines 3–4 of the manuscript published in AMTD: "Every 15 days, the CATS team computes 30-day averages of the CALIOP 532 nm scattering ratios between 22 and 28 km."

Why are only 4 months plotted in Fig. 2? What scattering ratios are used for the months not plotted? More specifics are need here.

The four months were chosen to illustrate the seasonal variability of these scattering ratios and thereby indicate that this seasonal variability is properly accounted for in the CATS calibrations.

Additionally, if zonal means are used, I would recommend putting standard deviations on the curves in Fig. 2 and discussing the amount of variability that is neglected by using mean values (this would need to be included in the error analysis as well).

The standard deviations of the 532 nm mean scattering ratios are quite small; for the January data shown in the paper (left panel of figure 2), the relative uncertainties (i.e., the standard error divided by the mean) for the 2° averages are all less than 0.2% (because the number of samples included in the averages is very large).

For Eqs. (2) and (3): why not just use the CALIOP 1064nm scattering ratios directly instead of assuming a backscatter color ratio?

Per our earlier response, the CALIOP 1064 nm SNR is too low to retrieve reliable estimates of high-altitude aerosol scattering ratios.

As I mentioned above, the uncertainty in the assumed backscatter color ratio is likely larger than just using 1064nm CALIOP data directly. Plus, the error in 1064nm CALIOP scattering ratios has already been characterized (*Vaughan et al.*, 2019) which would make the authors' uncertainty analysis more straightforward. Limb sounding instruments will, by far, give the highest accuracy scattering ratios at these altitudes. Did the authors explore any other alternatives to using CALIOP

for getting the aerosol scattering ratios? At the very least, the CALIOP scattering ratios should be compared to the climatology of SAGE II, SAGE III, GOMOS, etc...

Limb sounding instruments do not provide scattering ratios since they don't measure aerosol backscatter, only extinction coefficient. While we could use extinction measurements and assume a lidar ratio, none of the instruments operational at the time of CATS launch provided measurements near the CATS 1064 nm wavelength. SAGE III started operating in June 2017, but data wasn't available until roughly a month before CATS stopped operating. Using SAGE III 1022 nm extinction measurements as a comparable to the CALIOP 532 nm scattering ratio method would make a wonderful follow-on study, but is far out of the scope of the current paper.

**Don't oversell the approach**
I've touched on this throughout my comments above. There are several statements throughout the paper that are misleading considering the large uncertainty in having a significant amount of aerosols in the calibration region and the reliance on CALIOP to account for this. The authors state in the abstract that "Overall, CATS has demonstrated that direct calibration of the 1064nm channel is possible from a space based lidar using the molecular normalization technique". But this statement is only half true because the CATS calibration relies on having another, already calibrated, lidar in space (CALIOP). You can't characterize this as a "direct calibration" if 30–50% of your calibration (i.e. Fig. 2) relies on inter-calibrating to CALIOP! In essence, the authors follow a similar approach as has been done in previous work: derive a 1064nm calibration from calibrated 532nm backscatter.

CATS is the first space-based lidar to use the Rayleigh normalization technique to directly calibrate a 1064 nm backscatter channel. This is a true first and an important milestone for the backscatter lidar community. It should not be understated just because we have chosen to incorporate independent measurements to better quantify aerosol loading.

LITE, GLAS, and CALIOP did not have the SNR to directly calibrate using the Rayleigh normalization technique at 1064 nm. In the past, this technique has always been called the Rayleigh or molecular normalization technique. However, these previous applications were at shorter wavelengths (355 nm and 532 nm) and in altitude regions with small aerosol contributions. Thus, we understand the comments from one of our referees about calling it a Rayleigh normalization technique. We have changed the phrases "Rayleigh/molecular normalization technique" to "atmospheric normalization technique" and "Rayleigh profile" to "Rayleigh profile corrected for aerosol contributions". A sentence on page 4, lines 16-19 defines this name.

There are several instances of the authors being cagey about this. For example, on page 3 lines 30-31: "CATS exhibits high nighttime 1064 nm SNR, enabling 1064 nm attenuated total backscatter (ATB) direct calibration without any dependence on the CATS 532 nm signal". This statement is very misleading. Yes, you have no dependence on the CATS 532 nm signal, but you do depend on the CALIOP 532 nm signal and, on top of that, a CALIOP 532 nm signal that is conveniently already calibrated for you. If the CATS 532 nm signal was of sufficient quality, you would have certainty used the CATS 532 nm signal instead! Another example: page 13 lines 24-29. Here the

authors do say the aerosol loading is higher in the CATS calibration region than for CALIOP. Instead of waiting for the conclusion, the authors should mention this in the introduction and then again when discussing Fig. 2. Additionally, it is important to convey, quantitatively, the difference between the two: CALIOP has aerosol scattering ratios that are less than 1.02 in its calibration regions (Kar et al., 2018, their Figure 2b), CATS has values between 1.4-2.0 (Fig. 2). That is a very significant difference. In spite of all these difficulties, the general pathway the authors have taken to calibration CATS is likely the best approach. But, the authors need to chose their words carefully and convey that their approach is not going to be widely applicable to other 1064 nm lidars since they do not demonstrate an independent calibration of 1064 nm backscatter.

We have attempted to be upfront about the all of the limitations of our technique. For example, in the first paragraph of Section 2.1 we plainly state that "While this altitude region provides sufficient molecular scattering for the Rayleigh normalization technique, the aerosol loading in the lower stratosphere (22-26 km) is also higher than the 36-39 km region used to calibrate 532 nm CALIOP data. To improve the accuracy of the CATS nighttime calibration, the aerosol loading in the calibration region must be quantified, along with the ozone transmission profile, molecular backscatter profile, and polarization gain ratio (PGR)."

We do agree that choosing words carefully is important. We are reminded of this by the comment saying that "the CATS calibration relies on having another, already calibrated, lidar in space (CALIOP)", which is repeated in similar statements throughout this short comment. However, this assertion is simply not true; calibrating CATS does not rely on having another lidar in space. We could easily parameterize this aerosol loading based on a model (i.e., previous measurements and studies), a technique that is utilized routinely in the satellite remote sensing community, to calibrate CATS data with a larger uncertainty. Calibrating CATS 1064 nm data in the mid-stratosphere to < 10% uncertainties does rely on having another source of stratospheric aerosol loading measurements in space. Given the unanticipated demise of the CATS 532 nm channel, we were quite fortunate in being able to leverage the CALIOP data for this purpose.

Speculating on a way to truly do so is good fodder for the conclusion. The authors do some of this already, but I would encourage them to expand that discussion a bit. Have the authors considered the precision/accuracy trade offs between a lowered rep rate and increasing the altitude of the calibration range? If the CATS measurements weren't limited to between 51S-51N how would polar stratospheric clouds impact the calibration in polar regions?

A sentence was added to the second paragraph of the conclusion to elaborate on how decreasing the laser repetition rate of a future CATS-like backscatter lidar could provide a larger data frame, and thus a higher calibration altitude, based on the comments of both referees.

**More minor comments/edits**

The minor comments were taken into consideration and some of them were adapted in the latest version of the manuscript.